



# Linkage among Ice Crystal Microphysics, Mesoscale Dynamics and Cloud and Precipitation Structures Revealed by Collocated Microwave Radiometer and Multi-frequency Radar Observations

Jie Gong[1,2], Xiping Zeng[3], Dong L. Wu[2], S. Joseph Munchak[2], Xiaowen Li[4,2], Stefan Kneifel[5], Davide Ori[5], Liang Liao[4,2] and Donifan Barahona[2]

[1] Universities Space Research Association, Columbia, MD 21046, United States
[2] NASA Goddard Space Flight Center, Greenbelt, MD 20771, United States
[3] Army Research Laboratory, Adelphi, MD 20783, United States
[4] Morgan State University, Baltimore, MD 21251, United States
[5] University of Cologne, Cologne 50923, Germany

*Correspondence to*: Jie Gong (Jie.Gong@nasa.gov)

**Abstract.** Ice clouds and falling snow are ubiquitous globally and play important roles in the Earth's radiation budget and precipitation processes. Ice particle microphysical properties (e.g., size, habit and orientation) are not only influenced by ambient environment's dynamic and thermodynamic conditions, but also intimately connect to the cloud radiative effects and particle fall speeds, which therefore impact up to the future climate projection and down to the details of the surface precipitation (e.g., onset-time, location, type and strength).

Our previous work revealed that high-frequency Polarimetric radiance Difference (PD) from passive microwave sensors is a good indicator of the bulk aspect ratio of horizontally oriented ice particles that are often occur inside anvil clouds and/or stratiform precipitations. In this current work, we further investigate the dynamic/thermodynamic mechanisms and cloud/precipitation structures associated with ice-phase microphysics corresponding to different PD signals. In order to do so, collocated CloudSat radar (W-band) and Global Precipitation Measurement Dual-frequency Precipitation Radar (GPM-DPR, Ku/Ka bands) observations as well as European Centre for Medium-Range Weather Forecasts (ECMWF) atmosphere background profiles are grouped according to the magnitude of PD for only stratiform precipitation and/or anvil cloud scenes. We found that horizontally-oriented snow aggregates or large snow particles are likely the major contributor to the high-PD signals at 166 GHz, while low-PD magnitudes can be attributed to small cloud ice, randomly oriented snow aggregates, riming snow or super-cooled water. Further, high (low) PD scenes are found to be associated with stronger (weaker) wind shear and higher (lower) ambient humidity, both of which help promote (prohibit) the growth of frozen particles and the organization of convective systems. An ensemble of squall line cases is studied at the end to demonstrate that the PD asymmetry in the leading and trailing edges of the deep convection line is closely tied to the anvil cloud and stratiform precipitation layers respectively, suggesting the potential usefulness of PD as a proxy of stratiform/convective precipitation flag, as well as a proxy of convection life stage.



## 1 Introduction

Ice clouds and falling snow are ubiquitous. It is found that on average 50% of the surface precipitation globally is linked to ice
in clouds either through the production of snow from ice crystals, or through melting of ice into rain [Field and Heymsfield,
2015].  While the primary driver of precipitation amounts is determined by the amount of water vapor available to condense
and the forcing mechanism, ice microphysical processes play a key role in determining how much of that precipitation reaches
the ground and where. For example, ice fall speed is closely tied to the frozen particle habit (i.e., shape), size, density,
orientation, etc., and can hence influence the spatial and temporal distributions of precipitation [Milbrandt and Yau, 2006].
Studies also show that ice cloud radiative effect (CRE) is strongly dependent on ice microphysical properties [Liou et al., 2002;
Tang et al., 2017; Zeng et al., 2009a, 2009b]. Therefore, it is critically important to measure, understand and appropriately
incorporate ice microphysical properties in models in order to accurately capture the spatial and temporal variations of ice
cloud, falling snow and surface precipitation for the sake of improving weather prediction and climate projection.

Due to the complexity and multi-faceted characteristics of ice microphysics measurements, multi-frequency radar and
polarimetric radar are probably the best choices from the remote sensing point of view. For radar frequencies in millimeter to
sub-millimeter regimes whose wavelengths are comparable in size to large ice to precipitating particles, the strong scattering
signals and their amplitude differences at different wavelengths can provide ample information about vertical profiles of frozen
particle density, particle size distribution (PSD), phase, etc. [Kneifel et al., 2011; 2015; 2016; Kulie et al., 2014; Neto et al.,
2019]. Triple-frequency radar measurement techniques have been gaining attention and developed quickly in the past decade,
and have been implemented at some research ground stations or field campaigns. Polarimetric radar (or dual-polarization radar)
measures radio wave pulses that are horizontally and vertically polarized at the same frequency. Their reflectivity difference
($Z_{DR}$) and specific differential phase ($K_{DP}$) can help better constrain the retrieval of precipitation intensity and phase
discrimination. Doppler polarimetric radar, now implemented as NEXRAD (Next Generation Weather Radar) network over
the continental United States, gives further information of particle motion (e.g., final fall speed) and hence can help us better
determine storm strength and identify severe weather conditions such as tornado at the initiation stage [Simmons and Sutter,
2005].

Although active radar techniques are superb in revealing ice microphysical properties, their availability from space are limited
to a curtain or narrow swaths due to mass, power and data transfer requirements. Albeit many ongoing efforts are pushing
small-payload spaceborne radars such as RainCube Ka-band radar into space, their sensitivity, stability and reliability still
require further improvements. Currently in space only CloudSat W-band radar (and EarthCare W-band radar as a successor to
be launched in the near future) and Global Precipitation Measurement (GPM) Dual-frequency Precipitation Radar (DPR, Ku
and Ka-bands) are operating on a regular basis, unprecedentedly deepening our understanding of ice microphysical
characteristics globally [Stephens et al., 2002; Luo et al., 2008; Gettelman et al., 2010; Skofronick-Jackson et al., 2018].
However, their spatial and temporal coverages are still extremely limited (see instrument specifications in Section 2). Besides,
the near-nadir views are not conducive for polarimetric measurements of ice properties.





Satellite-borne passive microwave sensors with channels suitable for measuring hydrometers have been continuously monitoring Earth's weather for more than 30 years [Manaster et al., 2017]. High-frequency microwave channel (frequency > 150 GHz) signals are dominated by frozen particle scattering when relatively thick ice clouds or frozen precipitation are present along the line of sight, and are hence ideal for retrieving bulk ice cloud properties such as ice water path (IWP) [Gong and Wu, 70 2014; Eriksson et al., 2015]. Although not available on geostationary platforms, currently there are more than 10 healthy operating polar-orbiting or procession-orbiting passive microwave sensors with channels beyond 150 GHz [GPM ATBD, 2017]. With swath width typically over 1000 km and footprint size 7-15 km, the combination of using them can readily generate ice hydrometer products on temporal and spatial scales that suits the needs from both weather and climate studies.

Physical-based retrieval algorithms prevailed in hydrometer retrieval algorithms for passive microwave sensors. In these 75 algorithms, ice cloud profiles are estimated by accounting for the radiative transfer through frozen or liquid hydrometers as well as gas absorbers before reaching the complicated surfaces with a wide range of emissivity [e.g., Wu and Jiang, 2004 for Microwave Limb Sounder; NOAA MIRS 1D-Var retrieval system]. While some of the recent products have advanced from using spherical ice models to more realistic habits, random orientation is still nearly always assumed to reduce computational complexity and in these retrieval algorithms to decrease the degree of freedom for the otherwise even more severely under- 80 constrained inversion problem. Ancillary data such as temperature profile are often needed as well. To overcome these shortcomings, efforts have been putting forward to use spaceborne active sensor information to help improve or constrain the errorbar of the ice hydrometer retrieval products from passive microwave sensors [e.g., Evans et al., 2012; Gong and Wu, 2014]. In particular, the GPM team uses DPR retrieved hydrometer vertical profiles as either the apriori database or "training" datasets to generate their official passive microwave and joint retrieval products [Kummerrow et al., 2018; Turk et al., 2018].

Among currently available spaceborne high-frequency microwave sensors, GPM Microwave Imager (GPM-GMI) has a unique vertically-polarized (V-pol) and horizontally-polarized (H-pol) channel pair at 166 GHz. Gong and Wu [2017] found that the magnitude of 166 GHz polarized difference (PD), defined as the brightness temperature (TB) difference between V-pol and H-pol ($PD \equiv TB_V - TB_H$), is a good indicator of the presence of oriented ice particles. The largest PDs are found in mediumly cold TB, corresponding to predominately horizontally oriented ice or snow particles inside medium thick ice cloud (e.g., anvils) 90 or stratiform precipitation layer. This feature was also identified from 87 GHz TMI measurements [Prigent et al., 2005] and 157 GHz MADRAS measurements [Defer et al., 2014] before. PD almost diminishes for clear-sky and deep convective scenes. For the former, 166 GHz is proved to not sensitive to surface polarization except in the polar area when atmosphere is extremely dry [Zeng et al., 2019]. As for the latter, Gong and Wu [2017] provided several possible explanations, including random orientation of ice particles induced by turbulent environment inside deep convective cores, large irregular-shaped graupel, or 95 both V-pol and H-pol reach saturation at the same optical depth. Gong et al. [2017] further found that PD has a strong diurnal cycle over tropical land that is opposite to the diurnal cycle of cloud thickness and surface precipitation rate. The diurnal cycle of PD leads the latter two by ~ 2 hours, indicating that ice microphysics change over the convection life cycle is important to the final precipitation received at the ground. Nevertheless, all of the aforementioned papers studied passive sensor signals only. Scattering signals from passive sensors have very limited information on the vertical distribution of ice particles, and



hence did not answer some of the fundamental questions: **which altitude does PD information come from? What microphysical and environmental factors affect the observed PD variation over time and space? Can PD give more information in a broader context rather than just microphysics?** In this paper, these questions will be addressed by utilizing collocated GMI, DPR and CloudSat radar measurements as well as auxiliary environment settings.

This paper is organized as follows. Section 2 will introduce the dataset and methodology we use to make the composites of

climatology. We will present in Section 3 the differences of radar reflectivity, temperature and water vapor between high- and low-PD scenes. In Section 4, we will thoroughly discuss the underlying physical and microphysical mechanisms as well as consequences of such discrepancies. In Section 5, an ensemble of 46 squall line cases will be presented to showcase the potential use of high-frequency passive microwave PD observations to differentiate precipitation system life stage. Section 6 summarizes the whole work and point out several future study directions.

**2.Datasets and methodology**

**2.1 GPM core satellite and definition of PD regimes**

The Global Precipitation Measurement (GPM) mission core satellite, launched on February 27, 2014, carries a Dual-Frequency Precipitation Radar (DPR) and a Microwave Imager (GMI). DPR is composed of a Ku-band radar (KuPR) and a Ka-band radar (KaPR), making measurement at 13.6 GHz and 35.5 GHz, respectively. DPR takes cross-track scan with a footprint size

of $\sim 5 \times 5\ km^2$ at nadir and a swath width of 245 km for KuPR and 120 km for KaPR, respectively. Both KuPR and KaPR shoot 49 beams in each scan with a range resolution of 250 m (over-sampled to 125 m), but 25 KaPR beams are matched with KuPR footprints for the dual-frequency algorithm to work, and the rest 24 beams are in interlaced mode with range resolution of 250 m. Therefore, there are total three modes of DPR scanning pattern: normal scan by KuPR (NS), matched scan by KaPR (MS) and high-resolution interlaced scan by KaPR (HS)[1]. In this paper, we will mainly use KuPR measurements, and the

central-25 MS measurement is used whenever "KaPR" is mentioned. 2A.GPM.DPR Version 05A measured reflectivity without correction is used in this study. GPM core satellite flies at an altitude of 407 km in a precessing orbit covering the Earth's $65°S$ to $65°N$.

GMI is a 13-channel conical-scan microwave radiometer that sweeps the forward-looking cone at 48.5° (Earth incident angle of 52.8°). The channel frequency ranges from 10 to 183 GHz, and only 166 GHz V-pol and H-pol measured brightness

temperature (1B.GPM.GMI, Version 05A) will be studied in the current paper. 166 GHz footprint size is $7.2 \times 4.2\ km^2$ (cross-track and along-track, respectively), which projects as a swath width of 904 km on the Earth's surface in the cross-track direction with 221 pixels in each scan, the center part of which overlays with DPR scan for $\sim$ 67 seconds during each GMI scan (https://pmm.nasa.gov/gpm/flight-project/gmi).

---

[1] Prior to March 2018, the remaining 24 KaPR beams were interlaced at reduced vertical resolution but higher sensitivity to provide improved spatial sampling, but have since been matched to the outer swath KuPR to provide dual-frequency retrievals in the full DPR swath.



Gong and Wu [2017] constructed the two-dimensional probability density function (PDF) for PD-TB relationship for different
latitude ranges, one example is shown in Fig. 1 for the deep tropics ($5°S - Equator$). PD has a large spread when TB is in the
middle of the observed range, implying different cloud and precipitation regimes are likely embedded in this medium cold TB
regime, which would be impossible to separate if TB is the only metric to consider. For simplicity, we arbitrarily define four
regimes in Fig. 1: Regime #1 ($TB < 150K, PD < 5\,K$) represents deep convective scenes (called "Deep Convective Regime"
hereafter); Regime #2, #3 and #4 share the same TB bounds ($150K < TB < 230K$), but different PD ranges, namely "low-
PD" ($PD < 5\,K$), "medium-PD" ($5K < PD < 15\,K$) and "high-PD" ($PD > 15\,K$) regimes. This paper will focus on the
differences between "low-PD" and "high-PD" regimes, as one can imagine situations falling in "medium-PD" regime must be
in transition status between the "low-PD" and "high-PD" scenarios. Since the general characteristics of the PD-TB relationship
is largely latitude-independent, this four-regime definition can be applied globally to all GMI measurements, except there are
much less observations falling into the deep convective regime at high latitudes. Besides, some of the shallow convections that
are not as thick as deep convective cloud (e.g., congestus) may be wrongly classified into the "low-PD" regime. As convective
core area is much smaller than stratiform precipitation areas [e.g., Olson et al., 2001], the blend-in of shallow convection
structures should have negligible impact on the general statistics.

## 2.2 CloudSat radar and auxiliary datasets

The CloudSat mission, launched on April 28, 2006 to a 705 km altitude Sun-synchronized orbit, carries the Cloud Profiling
Radar (CPR). CPR is a nadir-looking W-band (94 GHz) radar with range resolution of 240 m and footprint size of
$1.4 \times 1.7 km^2$. The measured reflectivity vertical profiles from 2B-GEOPROF Version R05 product is used in this study.
As radar frequency increases from Ku-, Ka- to W-band, the radar sensitivity window also switches from precipitation to cloud.
The CPR reflectivities are subject to strong attenuation from rain and multiple scattering from large precipitation particles (rain
or ice). This becomes a serious issue in the range bins filled with heavy precipitation (i.e., from the melting layer to the ground).
Due to complicated melting process within the melting layer, which often shows as a layer of enhancement of radar reflectivity
(so-called "bright band"), as well as the liquid attenuation issue, we will avoid discussing any reflectivity signals below 5 km
(rough height of melting layer in the tropics) for all three radars throughout the paper. Water vapor throughout the profile can
also attenuate the reflectivity signal by up to 5 dBZ for CPR [Marchand and Mace, 2018], but we still use measured reflectivity
to avoid introducing additional assumptions that might complicate our analysis. The impact of water vapor attenuation at W-
band will be touched later in the discussion.
ECMWF-AUX Version R05 dataset produced by CloudSat team provides us auxiliary meteorological fields that are spatially
and temporally interpolated to CloudSat range resolution volumes from ECMWF high-resolution analysis [Partain, 2007].
Temperature, water vapor and horizontal wind profiles are compared for different PD scenarios.



### 2.3 Collocation of radar and passive imager footprints – match and mismatch

CloudSat-GPM Coincidence dataset Version 3B is a collection of collocated and coincident GMI, DPR and CPR measurements, which can be conveniently used for our current study. Details of collocation criteria and procedures can be found in Turk [2017]. This dataset has been used by many other researchers. For example, Yin et al. [2017] used collocated CPR-DPR reflectivity profiles from this dataset to study discrepancies of triple frequency radar signature and the inferred different microphysics processes between convective and stratiform regimes. In our study, we used more than three years of

data (March 2014 – October 2017) to produce a total of 3040 coincident observations globally. This number of samples is based on GMI footprint; as DPR and CPR footprint sizes are smaller, we first averaged multiple DPR and CPR profiles to one collocated GMI footprint, and then group the averaged reflectivity, temperature, water vapor, zonal and meridional wind profiles into four regimes according to the PD-TB values. Sample size separated in different categories can be found in Table 1.

Imperfect matching due to differences in footprint size, view-angle (i.e., atmospheric volume along line-of-sight is different even when the sight lines intersect), time or other factors can distort the compiled statistics. In our case, footprint size and line of sight mismatch are likely the largest sources of bias/uncertainty due to imperfect match. On one hand, CPR footprint is much smaller than DPR's and GMI's footprints, and therefore, any cloud/precipitation inhomogeneity in the scale smaller than ~5 km can result in discrepancies that are hard to evaluate. On the other hand, since match-up is defined to happen whenever

CPR beam intercepts with DPR beam at any altitude and at any DPR view-angle, the line-of-sight volume is quite different when DPR is at an off-nadir view-angle, and this problem is even more severe for GMI which always views at a slant angle. Even though a cosine function is multiplied to slightly mitigate this issue [Turk, 2017], 3D cloud inhomogeneity and beam-filling effects are again the culprit of uncertainty that is hard to justify. These two problems, however, are expected to be not too serious for our current study, because cloud inhomogeneity inside anvil and stratiform clouds is not as large as in deep

convective scenes. Nevertheless, we know they will increase the uncertainty of our results, and temporal difference (allowable to be up to 15 minutes) has a similar impact. Only footprint mismatch might add an extra bias though, as will be discussed in Section 3.1.

As this coincident dataset does not contain collocated wind and bright band information from DPR, collocated indices are matched back to CloudSat ECMWF-AUX and 2A.GPM.DPR data files to extract the wind and bright band height/width

information.

### 3. Differences between high-PD and low-PD scenes

### 3.1 Radar reflectivity differences between high-PD and low-PD scenes

Using 3.5 years of collocated radar reflectivity profiles, we can composite the two-dimensional probability density function (2D-PDF) respectively from CloudSat (color shaded) and KuPR (color contours) for the four regimes for the tropics, which is



shown in Fig. 2. CloudSat's 2D-PDF separates "deep convective" scenario clearly from the rest three scenarios by having no bright band kink at ~ 5 km, great amount of high clouds, and the center of highest occurrence of reflectivity located in the middle-upper troposphere (7-12 km) at around 15 dBZ. The PDF of "Low-PD" scenario is the closest to that of the "deep convective" scenario among the remaining three. Compared with "high-PD" scenario, the "low-PD" one apparently has more high clouds that are thinner than those from deep convective scenes as the reflectivity magnitudes are smaller. As PD becomes

larger, the bright band kink at ~ 5 km becomes more and more distinguished while the maximum occurrence of reflectivity also shifts down toward middle troposphere (5-8 km). This indicates the scene is more and more stratiform precipitation-alike when PD magnitude increases. For KuPR's CFAD, as Ku-band is only sensitive to the precipitation-sized particle, we basically observe the same story as with CloudSat's 2D-PDF, except KuPR cannot see high altitude anvil clouds. Because the KuPR reflectivity does not saturate with particle size as rapidly as Cloudsat, we can also infer large ice particles high in the atmosphere

in the deep convective and low PD cases, while the strong increase in reflectivity towards the bright band in the high-PD case is indicative of aggregates.

The 1D-views of mean reflectivity profile from CloudSat, KaPR and KuPR ingeminate the proceeding story in a more clear and concise way, as shown in Fig. 3. Basically, Fig. 3 is the PDF-weighted mean along the x-axis of Fig. 2. Since 2A.GPM.DPR dataset also reports the altitude of bright band (i.e., melting layer), Fig. 3b and 3c are plotted against altitude with respect to

the melting level. As we stated in Section 2, we do not intend to discuss any signals below the melting layer since CloudSat reflectivity is likely strongly attenuated below the melting layer, and measured reflectivity is used for all three radars without any attenuation correction. Above the melting layer, high-level cloud (> 9 km) is less and less present while middle level cloud is more and more thick (5-8 km) when cloud regime switches from #1 "deep convective" (dark blue) to #4 "high-PD" (red). If we check the KaPR and KuPR profiles in Fig. 3b and 3c, however, we see roughly two distinct regimes: one includes

scenario#1 and #2 ("deep convective" and "low PD") that has more precipitation-sized particles throughout the upper-middle troposphere, suggesting the convection and related cloud are still actively present within the column. On the contrary, the other regime, including scenario#3 and #4 ("medium" and "high" PDs), comprises much less precipitation-sized particles up loft until close to the top of the melting layer, where the sharp enhancement of reflectivity indicates fast and efficient growth from small ice particles to large fluffy snow aggregates. This is reasonable to happen microphysically because the sticking efficiency

of two-ice-crystal collision increases rapidly near the melting layer. The latter scenario indicates the late stage of convection life cycle where development of new cloud and convective cell disappears and a stable stratiform layer dominates the whole column.

Based on Fig. 2 and Fig. 3, we can conclude for the discrepancies between high- and low-PD scenarios based on pure radar observations: "low-PD" scenario has more high cloud and large ice particles high into the troposphere, more "development

stage"-like, while "high-PD" scenario has much less high cloud but more middle-level cloud, with snow aggregation evident near the top of the melting layer. Therefore, "high-PD" scenario shows a distinct bright band, or melting layer signature, which is more "stratiform"-like or "decaying stage"-like. At this point, however, we cannot yet conclude whether it's the preferably horizontally oriented large snow aggregates above the melting layer that cause the "high-PD" signals, or it's the randomly





oriented ice/snow particles in the upper-middle troposphere that effectively damps the PD signal in the "low-PD" scenario
from just looking at single radar composite. We will discuss each of these possibilities in conjunction with radiative transfer
model simulation results in Section 4.

## 3.2 Background atmosphere differences between high-PD and low-PD scenes

In this subsection, collocation cases from tropics, mid-latitude winter and summer are averaged separately considering they
locate in different weather regimes. Because CloudSat CPR is only operated in daytime mode after 2011, collocation samples
over Southern Hemisphere (SH) are very sparse (Table 1), and therefore only Northern Hemisphere (NH) winter and summer
situations are shown in Fig. 4 for temperature and water vapor, and Fig. 5 for zonal and meridional winds. The definitions of
extended winter (November – March) and summer (May – September) are used in order to enlarge the sample sizes. Although
ECMWF-AUX, extracted from ECMWF high-resolution analysis, cannot be considered as "observation", it is proven to have
high quality in capturing the mesoscale to large scale variations of atmospheric fields [Burgess et al., 2013; Gong et al., 2015].
Nevertheless, due to imperfect collocation as we discussed in Section 2.3 and the fact that convections are not perfectly
represented by analysis, we would expect the ECMWF-AUX data represent both the ambient and in-cloud dynamic and
thermodynamic conditions, which is unfortunately inseparable.

High and low PD regimes do not differentiate from each other clearly on the background temperature profiles as shown in Fig.
4 except for boreal winter when "high-PD" scenes are on average ~ 10K warmer than the "low-PD" scenes throughout the
troposphere. Water vapor amount, however, is consistently higher for "high-PD" scenes than for "low-PD" scenes in both
tropics and mid-latitude, which translates into higher relative humidity when temperature profiles are almost identical. In the
tropics and boreal summer, widespread anvil clouds and stratiform precipitation are mostly tied to deep convective systems
such as mesoscale convective system (MCS), hurricane, squall line, etc. Based on the consistent water vapor discrepancy
between the two regimes, we can assert that higher humidity creates a more favorable environmental condition to promote a
higher 166 GHz PD signal. This feature in-cloud can be understood as higher humidity usually boosts up the deposition growth
of ice particle in size at non-convective regime. Given the same orientation distribution, larger-sized particle will result in
stronger 166 GHz PD, as shown in recent simulations in Brath et al. [2019]. Considering this difference for ambient
environment, higher humidity also tends to spawn more vigorous convective systems or more organized convection, both of
which produce vast areas of stratiform precipitation and therefore a higher 166 GHz signal.

The wind and wind shear differences between the two scenarios are less conclusive, which are not statistically significantly
differentiated from one another except for the meridional wind in boreal winter (middle panels of Fig. 5). But we can see a
general pattern that both zonal and meridional low-level shears for "high-PD" scenes are higher than those for the "low-PD"
scenes, which in general promotes organized convection (MCSs) with large stratiform precipitation areas. In the tropics, there
is barely any wind shear for "low-PD" scenes, but "high-PD" scene exhibits stronger meridional southerly shear in the lower-
level, and stronger zonal westerly shear in the upper-level. This fits into the conceptual model of mesoscale convective system
(MCS) with a rear-heavy deck in that the lower-level rear inflow jet is nearly perpendicular to upper-level front-to-rear flow



[Markowski and Richardson, 2010]. Weak shear in "low-PD" scenes, on the contrary, may indicate that the associated convection is either isolated, mature or dissipated so stratiform deck area is either very small or not present. For boreal summer (bottom panels of Fig. 5), the stronger lower-level wind shear in "high-PD" scenes likely promote more organized convective systems or organized convective vortices, as have been shown in many modeling and observational studies [e.g., Houze, 2004; Chen et al., 2015], so "high-PD" scenes are easier to occur if they are caused by large horizontally-aligned snow aggregates that are often observed in the stratiform precipitation region. We will show in Section 5 that this hypothesis is valid for the quasi-linear MCS – Squall line situation.

In boreal winter, convection and stratiform precipitation are most frequently associated with frontal systems. The distinguished differences in temperature (~ 10K), humidity (much wetter for "high-PD" scenes) and wind (northwest wind for "high-PD" scenes versus southwest wind for "low-PD" scenes) all strongly indicate that the two scenarios correspond to two very different weather regimes. "High-PD" environment fits the warm front setting while "low-PD" environment fits the cold front configuration. We will not inspect further into this speculation however, due to the page limit, and will leave this as part of the future work.

## 4. Possible causes of PD differences

We presented evidences in Section 3 that the "high-PD" scenario is tied to a quick growth of ice particles into horizontally aligned snow particles in the middle troposphere above the melting layer. This growth process can be aggregation or water vapor/liquid deposition. However, for the "low-PD" scenario, we cannot differentiate from single radar reflectivity profiles whether it's high-altitude randomly oriented large snow particles that effectively damps the PD signal, or the clouds are dominated by small cloud ice particles that 166 GHz PD is not very sensitive to even if the particles are preferably horizontally oriented, or some other possibilities. Therefore, in this section, we will try to delineate some of the unique microphysical characteristics that exhibit under "low-PD" scenario and discuss why they lead to the small PD signals at 166 GHz. This investigation will not only help us understand our particular problem more deeply, but also help explore the potential usability of PD when active radar is not present, which is most of the cases.

As briefly introduced in Section 1, triple frequency radar reflectivity differences can reveal many quantitative structures of ice microphysical properties. In particular, Mason et al. [2019] has demonstrated that PSD shape parameter and ice morphology are the leading two factors that contribute to the spread of reflectivity differences on the triple frequency diagram, while density and axial ratio play a secondary role. However, unlike the RTM calculation and ground triple frequency radar measurements that have dedicated perfect-match design, our imperfect matches between CPR and DPR due to temporal, footprint size and viewing geometry differences will confine our analysis here only to a qualitative level. But since DPR-Ku and Ka are matched, direct comparison against RTM simulations are possible, as will be shown in Fig. 7 and related context.

Dual-Frequency Ratio (DFR) or Dual-Wavelength Ratio (DWR) between each of the two pairs of radar measurement are defined as reflectivity difference when they are in dBZ units, for example, $DFR_{Ku/Ka} \equiv \frac{Reflectivity_{Ku}}{Reflectivity_{Ka}}[mm^6 m^{-3}] \equiv Z_{ku} -$





$Z_{ka}$ [dBZ]. Two-dimensional PDFs from "high-PD" (blue contours) and "low-PD" (colored shades) scenarios using all
collocated samples between 5.5 – 15 km altitude range are presented on the DFR diagram in Fig. 6. Measurements below 5.5
km are excluded to avoid impacts from melting layer and potential saturation below. Because multiple radar profiles are
averaged to a single GMI footprint, sensitivity thresholds changed accordingly. Yin et al. [2017]'s thresholds are used here to
exclude any reflectivity signals below, which are 13, 11 and 2 dB for DPR-Ku, DPR-Ka and CPR. DFRs for two ice particle
shapes (dendrite aggregate and graupel spheroid) and two riming conditions (unrimed, and heavily rimed with effective liquid
water path of $2 kg/m^2$) are calculated and overlaid on Fig. 6b as rough references of theoretical calculated value, which are
arbitrarily moved to the right by 5 dBZ in order to match the center of the observed PDFs. See Leinonen and Szyrmer [2015]
for details of the RTM, scattering property and ice morphology definitions.

$DFR_{Ku/Ka}$ does not show a distinguished difference between the two scenarios, but we can see a hint of slight tendency toward
larger $DFR_{Ku/Ka}$ values for "high-PD" scenario, which suggests that bigger fluffy snow aggregates are the cause of greater
Ku and Ka differences. As for $DFR_{Ku/W}$ (Fig. 6a) and $DFR_{Ka/W}$ (Fig. 6b), the largest power does not center around 0 dB but
at 5 dB instead, which is likely due to the different minimum detectable reflectivity for CPR and DPR [Skonfronick-Jackson
et al., 2019]. This is also observed by Yin et al. [2017] for deep convective and stratiform precipitation scenes using collocated
CPR-DPR measurements. Other than the biggest blob of power centered at 5 dB, which basically suggests that there is little
difference in Ku-W and Ka-W response for most of the low-PD samples considering the disparity in their detection thresholds,
there are some samples that have very large $DFR_{Ku/W}$ and $DFR_{Ka/W}$ values. These samples line up closer with the theoretical
calculation for heavily rimed graupels, indicating that riming might be the cause for the low-PD signal for these samples. The
two modes roughly separate at ~ 12 dB. This "double-mode" feature in $DFR_{Ku/W}$ and $DFR_{Ka/W}$ strongly suggest that "low-
PD" scenarios have at least two major microphysical contributors.

To further investigate the cause of "double-mode" feature in $DFR_{Ku/W}$ and $DFR_{Ka/W}$, collocated temperature soundings at
the same height level of radar reflectivity measurements are used to sort out the $DFR_{Ku/Ka}$ (Fig. 7a) and $DFR_{Ku/W}$ (Fig. 7b),
and we further separate the "low-PD" scenario into four sub-categories: (1) large $DFR_{Ka/W}$ (> 12 dB), cold temperature ($T <
-20°C$); (2) large $DFR_{Ka/W}$ (>12 dB), warm temperature ($T > -20°C$); (3) small $DFR_{Ka/W}$ (< 12 dB), cold temperature ($T <
-20°C$); and (4) small $DFR_{Ka/W}$ (< 12 dB), warm temperature ($T > -20°C$). The temperature threshold ($-20°C$) roughly
separates ice particles that are likely unrimed and may be partially rimed, respectively. Dias Neto et al. [2019] also found ice
PSD widens toward larger DFR in the $-20°C < T < -10°C$ range, indicating that $-20°C$ is a good threshold to use. The
"high-PD" samples are shown as red asteroids in Fig. 7. For easy reading, these four sub-categories are named as "PD$_{Low}$-
DFR$_{Large}$-T$_{Cold}$" (light green asteroid), "PD$_{Low}$-DFR$_{Large}$-T$_{Warm}$" (dark green asteroid), "PD$_{Low}$-DFR$_{Small}$-T$_{Cold}$" (black dot),
"PD$_{Low}$-DFR$_{Small}$-T$_{Warm}$" (blue asteroid), respectively.

Based on Fig. 7a and 7b, we can first conclude that all "high-PD" scenes (red asteroid) are within relatively warm temperature
range ($-20°C < T < 0°C$), which locates in the middle-low troposphere if we use height coordinate (not shown) and are hence
likely consist of aggregated snow particles rather than ice cloud particles. About 50% of the "low-PD" measurements are





located in the low temperature, low DFR regime (black dots). These are very likely ice cloud particles that are small and hence introduce very weak $DFR_{Ku/Ka}$, and only marginally detectible by CPR considering averaging of multiple CPR profiles onto GMI footprint significantly degrades the lower boundary of its sensitivity threshold. The rest "low-PD" observations (blue,

light green and dark green asteroids) are inherently indistinguishable from "high-PD" ones on the $DFR_{Ku/Ka}$ axis (Fig. 7a), which indicates they are about the same size with those snow particles that generate the "high-PD" signal. However, scenes with larger $DFR_{Ka/W}$ (green asteroid) show a significantly larger $DFR_{Ku/W}$ signal too, as shown in Fig. 7b. Comparison between Fig. 7a and 7b suggests that there must be certain mechanism(s) for these large $DFR_{Ku/W}$ (or $DFR_{Ka/W}$) scenes that effectively decrease the W-band radar reflectivity. These mechanisms could be particle riming, as suggested by the RTM

calculations in Fig. 6b, super-cooled liquid water that works effectively as a W-band reflectivity damper as well as 166 GHz PD damper [Xie et al., 2017], or water vapor that also impacts the W-band much more seriously than Ku or Ka bands. According to Fig. 4, "high-PD" scenes unanimously have higher water vapor amount than "low-PD" scenes. Therefore, water vapor abundance could serve as a plausible explanation to explain the small $DFR_{Ku/W}$ for "high-PD" scenes (red asteroids in Fig. 7), but not for "low-PD" scenes. Therefore, frozen particle riming and super-cooled liquid water are the two most plausible

candidates to explain the behaviors of those green asteroids (PD_Low-DFR_Large). For PD_Low-DFR_Small scenes (black dot and blue asteroids), we have discussed that black dot scenes probably correspond to cloud ice particles because of cold temperature and relatively high altitude where they are located, and the blue asteroids scenes may be explained as randomly oriented unrimed snow aggregates.

Particle density isolines overlaid on DFR-Z plots can help further delineate the different microphysical regimes that "high-

PD" and "low-PD" scenarios fall into, as shown in Fig. 7c and 7d [Liao and Meneghini, 2011]. Keep in mind that Fig. 7c is comparable quantitatively to the theoretical isolines (other than that the water vapor is not corrected) as Ku-Ka are perfectly matched in beam, while 7d can only assist our interpretation qualitatively. The "high-PD" scenes are separated from the three "low-PD" sub-categories in Fig. 7c, and more clearly in Fig. 7d (PD_Low-DFR_Small-T_cold samples, the black dots in Fig. 7a and 7b, are not shown because they are likely from cloud ice particles). In particular, one can see that given the same small DFR,

the blue asteroids are generally denser than red asteroids, which agrees with our earlier hypothesis that "high-PD" is mainly induced by fluffy snow aggregates where ice crystals are loosely attached to each other during the rapid aggregation process, and therefore tends to fall slowly and orient horizontally because of the large geometric area, while some of the "low-PD" signals (blue asteroids) are induced by denser snow aggregates that are comparable in effective diameter with those ones in the "high-PD" scenes, but because the ice crystals are more tightly collided together in these snow aggregates, they tend to not

have a preferred shape nor orientation. The dark green asteroids, are suggested by Fig. 7c to be even denser (except a few outliers on the top-right corner), which strongly suggest that they are possibly experiencing riming. Fig. 7d tells contradictory story about these green asteroids, however. But note that observations in Fig. 7d are not directly comparable with RTM calculations because of imperfect match, we focus on and trust more of results presented in Fig. 7c (so does Fig. 7a).



To summarize this section, we are able to provide more evidences that "high-PD" signal is mainly induced by horizontally
oriented fluffy snow aggregates, while also successfully delineating several possible microphysical mechanisms of "low-PD"
signal, which are: (1) small cloud ice particles; (2) more densely aggregated snow particles that tend to orient randomly; (3)
riming snow; (4) super-cooled liquid water. This section demonstrates the value and importance of closely matched triple
frequency radar measurements in conjunction with accurate atmospheric soundings. We also find that 166 GHz PD indeed is
primarily sensitive to the orientation of large snow particles instead of small cloud ice particles. Adding polarization
measurements at a sub-mm frequency (e.g., 640 GHz from ICI mission) may likely aid on disentangling orientation of cloud
ice from that of snow particles for a passive sensor.

**5. PD variations through cloud-precipitation life stage: demonstration with squall lines**

Convective systems usually experience a complete life cycle of development. At the developing stage, the deep convective
tower(s) quickly shoots up and anvil deck quickly spreads out. At the decaying stage, convective core dissipates and is replaced
with a stable stratiform precipitating area. By far we know that anvil cloud is likely associated with low-PD while high-PD is
likely associated with a stratiform layer. Can we use PD reversely as a proxy to tell which stage a convective system is at? If
yes, does high-PD necessarily associate with more intense surface precipitation, while low-PD is on the contrary? In this
section, we will explore the potential usefulness of PD on a relatively simple natural testbed – squall line system. Squall line,
as a subset of typical convective system, is an ideal testbed because it is quasi-linear (i.e., no rotation like hurricane or frontal
system), and the anvil head in the leading edge and the vast area of stratiform precipitation layer in the trailing edge are easily
separable on GMI images.

46 squall line cases are selected manually from GPM swaths of observations (see Table A1 for detailed location, orbit number
and date. For these 46 cases, we require: (1) the line of deep convection must be captured by DPR; (2) the squall line moving
direction must be quasi-perpendicular to GPM orbit. The latter requirement assures that each GMI conical scan can slice
375   through the squall line whenever the collocated DPR captures a deep convective core. An example of the selected case is given
in Fig. 8, where the leading edge and trailing edge is to the left and right, respectively, because this case happened over Africa.
From Fig. 8b we can clearly visualize that PD is significantly higher behind the line of deep convection identified from Fig.
8c on the DPR image. To composite the ensemble of 46 cases together, we first identify the footprint location on each DPR
scan that has the maximum Ku-PR retrieved near-surface precipitation rate ($PR_{sfc}$), which is required to be beyond a threshold
380   ($PR_{sfc} > 25\ mm/h$). This threshold is arbitrarily set, but works robustly against a range of values ( $20 - 50\ mm/h$ are tested,
and only the number of qualified samples are changed without altering the conclusion). The location of identified deep
convective centers are shown in Fig. A1. Once the center of convection on a DPR scan is identified, the collocated GMI
footprint and its index on the scan are then pin-pointed. Next, the footprints to the leading edge and trailing edge are sorted
accordingly against the convective center. Lastly, all selected GMI and DPR scans are aligned together with GMI's deep
385   convective center footprint in the middle, trailing edge to the left and leading edge to the right. The averaged cross-section of




166 GHz TB and PD responses for each case are shown as black lines in Fig. 9a and 9b, respectively, while the ensemble means are the red bold lines. The DPR scan widths are shown as light grey rectangles in Fig. 9a and 9b for comparison.

Both TB and PD across the squall line are asymmetric about the deep convective center. The ensemble mean of TB is ~ 10 K colder to the trailing edge than to the leading edge, which translates into a radiatively thicker anvil cloud/falling snow layer
behind the squall line than ahead of it. The width of depression of TB is also much broader in the trailing edge than in the leading edge. Both features agree well with our conceptual picture of squall line system [Houze, 2004]. Cetrone and Houze [2011] selected and studied another ensemble of Africa squall lines, and they found that the geometric thicknesses of leading anvil and trailing stratiform cloud layers are not significantly different. So a radiative thickness difference of ~ 10 K in our ensemble case could indicate larger snow particles in the trailing edge than fresh anvil cloud ice particles in the leading edge.
The ensemble mean of PD, as expected, displays a double-peak in front and behind the deep convective center. The peak occurs ~ 70 km and ~ 25 km away from the center on the trailing edge and leading edge, respectively. The magnitude of PD in the trailing edge is apparently larger (7 K) and spread more wildly than that in the leading edge (5.5 K). Interestingly, PD in the deep convection is not zero but ~ 4.5 K. One can see from the general statistics in Fig. 1 that PD is ~ 4 K when TB is about 120 K, which are consistent with the mean TB and PD values we found for deep convective cores from our squall line
ensemble. We can further see from Fig. 10b when the distribution of PD is plotted that discernable number of convective pixels has large PD values and hence skewed the mean toward positive side. A 5.5 K peak PD value in the leading edge suggests that there are also significant amount of large horizontally-aligned ice particles inside leading anvil decks, which is consistent with previous findings in Cetrone and Houze [2011].

The $PR_{sfc} - PD$ relationship is further evaluated. We perform this evaluation from two perspectives. First, we would like to
check if across the squall line, precipitation intensity is significantly different or not, and whether it's related to the magnitude of PD. A threshold of 5 mm/h is used as a rough threshold to exclude convective rain scenes because the magnitude of PD inside convection involves too many complicated dynamic/microphysical processes and this is not what this paper is aiming for. Away from the convection, a correlation of 0.77 (significant at 99.9% confidence level) is achieved, as shown in Fig. 9c. $PR_{sfc}$ reaches 2 mm/h in the stratiform zone behind the deep convection, which is < 1 mm/h in the leading edge. Considering
that it takes some time for snow aggregates to fall down to the surface while squall line keeps on moving forward, a spatially-lagged correlation would be more physically meaningful [Gong et al., 2017]. Unfortunately, the DPR scan is too narrow for us to conduct such a lag-correlation test. The other way to evaluate the $PR_{sfc} - PD$ relationship is to composite the statistics. Since Level 2 DPR retrievals provide the precipitation type flag, which classify precipitation into three types: stratiform, convective and other, we use these three flags to separate our ensemble data into three categories, and composite the 2D PDF
of $PR_{sfc} - PD$ for each of them. Note that "stratiform precipitation" flag only occurs in the trailing edge, but "other precipitation" flag may include precipitation happens everywhere within the domain, so we add an extra criterion that only the footprints located at the leading edge are selected to use. The 2D PDF composites of $PR_{sfc} - PD$ for stratiform, deep convective and leading edge are shown in Fig. 10, which are completely different, confirming that the cloud-precipitation



process belongs to completely different regimes for each of the three types. The most likelihood of PD in stratiform region
reaches ~ 10 K, while in convective regime it centers around 4-6 K, and the value spreads out in the leading edge from as small
as 2 K to up to 10 K. PD is only positively correlated with surface precipitation rate in the stratiform region and only when PD
is high. A 2 K increase of PD from 8 K to 10 K corresponds to four times of $PR_{sfc}$ increase from 1 mm/h to 4 mm/h (note that
the horizontal axis in Fig. 10 is logarithmic), which is quite remarkable and again demonstrates that ice microphysics is so
important to the local precipitation intensity and its variation. In the convective regime, PD is weakly negatively correlated
with surface precipitation, but when PD is large (> 8K roughly), the correlation becomes negligible or even positive, which is
likely due to the fact that these pixels are close to the boundary of convective-stratiform separation. Keep in mind that these
correlations (Fig. 10) can only be performed within the narrow DPR swath (grey bars in Fig. 9a and 9b) where both DPR
identified precipitation flag and collocated 166 GHz PD measurements are available, while the majority of stratiform
precipitation and fresh anvil leading head are outside of the DPR swath using our current ensemble technique, so Fig. 10 is
just looking at a spot of the leopard, and more comprehensive works are needed in the future to recover the whole picture.
Although the steep slope means that $PR_{sfc}$ is very sensitive to the magnitude of PD in the stratiform region when PD is large,
the overall spread of the 2D PDF suggests that we cannot incautiously use PD as a new parameter for surface precipitation
retrieval. Rather, this finding suggests that 166 GHz PD could be and should be considered as an extra constraint to GMI-only
precipitation retrieval because of its added value in certain regimes, which remains as a worthwhile topic for future exploration.
Meanwhile, we can clearly see from Fig. 10c that in non-stratiform regime (i.e., leading edge in our squall line cases), PD is
not correlated with $PR_{sfc}$ at all. The contrast of 2D PDF shapes in the trailing edge versus in the leading edge again goes along
with our understanding that large horizontally aligned snow aggregates tend to occur in the stratiform layer and tend to
precipitate down in a short frame of time, while it takes longer time for those relatively small snowflakes hanging up in the
fresh anvil deck in the leading edge to fall to the ground, the latter of which apparently will enjoy varied experiences during
the falling process. Furthermore, PD from 166 GHz cannot be used solely to diagnose the life stage of convective system
either, but multiple spaciously coherent PD measurements spanning from MW to sub-mm or even infrared spectra may realize
this function eventually without any involvement of active remote sensing. For example, Olson et al. [2001] developed a
passive MW-only flag system, called "convective-stratiform index (CSI)", to flag out convective and stratiform scenes, which
relies on a combination use of TB and TB gradient at 19, 37 and 85 GHz, as well as 85 GHz PD magnitude. This was back in
TRMM era when 85 GHz is the highest available frequency making measurement at both V-pol and H-pol. Although not
shown nor discussed in the main text, Fig. A2 includes the same squall line ensemble of TB and PD at 89 GHz from GMI
measurements. We can see that 89 GHz PD spreads all over the place and does not show a good correspondence to deep
convective or stratiform zones. Based on our investigation in this paper, 166 GHz PD is likely a better candidate to update the
CSI algorithm, which is left for exploration in the future.



## 6. Conclusions and Future Works

At 166 GHz, GMI currently makes the highest frequency dual-polarized radiance measurements from space. Gong and Wu [2017] and Gong et al. [2017] thoroughly studied the PD and TB signals from GMI, and laid out several possible microphysical mechanisms that can explain the PD signal under different TB regimes. In this paper, leveraging on the collocated DPR and CPR radar reflectivity measurements, we are, for the first time, able to delineate the microphysical properties not only for high-PD scenes, but also for low-PD scenes for the same medium-cold TB regime. As high-frequency MW radiance measurements can only tell column integrated mass quantities such as ice water path for this medium-cold TB regime, the PD measurement allows us to diagnose much more information about the microphysical processes occurring in the profile. The analysis of PD and radar data in this paper suggests that 166 GHz PD is closely associated with horizontally oriented large fluffy snow aggregates in the stratiform precipitation layers that tends to melt and fall as precipitation soon. Low 166 GHz PD signals, however, are associated with more complicated situations. At least four mechanisms are found possibly responsible for the low PD signals: (1) small cloud ice particles up aloft that 166 GHz PD is barely sensitive to; (2) more densely aggregated snow particles that tend to orient randomly; (3) riming snow particles that effectively damp the PD signal; (4) super-cooled liquid water that also damps the PD signal. With only 166 GHz TB and PD observations, it is difficult to distinguish which mechanism dominates a single scene, but better diagnostic approaches can be developed in the future using adjacent passive-only observations in conjunction with accurate atmospheric background measurements or analysis. As sub-millimeter is more sensitive to smaller particles, multi-frequency PD measurements at MW and sub-mm spectra can be provided the best approach to separate particle size, shape and orientation information from cloud ice and snow aggregates, as already shown in the RTM simulations by Brath et al. [2019].

This paper also demonstrates the value of using collocated triple-frequency radar measurements to disentangle the complicated microphysical characteristics along the passive MW line of sight. However, due to beam-filling and other mis-match issues (e.g., footprint, view-angle and temporal disparities), our "pseudo-triple-frequency" radar (DPR+CPR) statistics are distorted and displaced, and therefore cannot be directly compared with RTM simulations quantitatively. Since perfectly matched triple-frequency radar space mission will be likely unavailable in the near future, using collocated ground or "pseudo-collocated" space radar measurements together with passive MW PD observations are probably the best approach to extend our knowledge about passive MW PD signals for a better scientific and/or operational use.

We lastly scrutinized 166 GHz PD and DPR behaviors in an ensemble of squall line cases. We found out that PD is larger in the trailing edge where large snow aggregates are predominant in the stratiform layer, while PD is smaller in the leading edge where fresh anvil decks dominate the scene. Surface precipitation, as expected, is positively correlated with PD in the stratiform region when high-PD occurs. Three possible ways of using PD are then discussed, which are: (1) using 166 GHz PD to constrain passive-only surface precipitation retrieval under certain conditions; (2) using multi-frequency PD and TB measurements to help diagnose the life stage of convective system; (3) improving the design of passive-only stratiform/convective precipitation flags. They will be explored in the future.





**Acknowledgement**

We are grateful to the dedicated experts from CloudSat and GPM teams who maintain and distribute the high scientific-quality
Level 1 and Level 2 data. We are in particular in debt to GPM precipitation feature product producer Dr. Chuntao Liu at Texas
A&M Corpse Christi. Discussions with Drs. Manfred Brath, Ian Adams and Chuntao Liu, among others, greatly help make
this paper a better shape. This work is mainly conducted under the support from NASA Grant# 80NSSC20K0087. Funding
supports from Grant#NNX16AM06G and NASA ROSES NNH18ZDA001N-RRNES are also highly appreciated.

**Author Contributions**

Dr. Gong and Dr. Munchak initiated the original idea. Dr. Gong conducted most of the data analysis and results interpretation.
Dr. Munchak provided part of the Ku/Ka CFAD composites. Dr. Zeng, Dr. Wu and Dr. Li were heavily involved in interpreting
the results. In addition, Dr. Li provided the squall line ensemble cases. Drs. Kneifel and Ori provided the theoretical
calculations of the triple frequency radar DFR and helped on explaining some of the observational property. Dr. Liang provided
the theoretical calculations of the density isolines. Dr. Barahona provided the global model perspective for this study.

**Data Distribution**


GPM data and collocated GPM-CPR data are available to public at NASA PPS FTP at https://pmm.nasa.gov/data-
access/downloads/gpm. CloudSat and ECMWF-AUX data are made available from CloudSat FTP at
http://www.cloudsat.cira.colostate.edu/order-data. High/low PD statistics can be provided to readers upon request.

**Competing Interests Disclaimer**

The authors claim no competing interests against any institutions or individuals.

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






| Region | Latitude bands | Season | Total # Samples | | High-PD | Mid-PD | Low-PD | Deep Convective |
|---|---|---|---|---|---|---|---|---|
| **Tropics** | $30°S - 30°N$ | All-year | Ocn[*3] | 779 | 16 | 100 | 644 | 19 |
| | | | Lnd | 320 | 0 | 86 | 227 | 7 |
| **NH** | $30°N - 50°N$ | NDJFM[*1] | Ocn | 336 | 20 | 303 | 13 | 0 |
| | | | Lnd | 970 | 0 | 465 | 505 | 0 |
| **NH** | $30°N - 50°N$ | MJJAS[*2] | Ocn | 107 | 15 | 85 | 7 | 0 |
| | | | Lnd | 153 | 1 | 121 | 28 | 3 |
| **SH** | $30°S - 50°S$ | NDJFM | Ocn | 110 | 2 | 106 | 2 | 0 |
| | | | Lnd | 10 | 0 | 10 | 0 | 0 |
| **SH** | $30°S - 50°S$ | MJJAS | Ocn | 181 | 7 | 170 | 3 | 1 |
| | | | Lnd | 74 | 1 | 70 | 3 | 0 |
| **Total** | | | | 3040 | 62 | 1516 | 1432 | 30 |

[*1]: November – March

[*2]: May - September

[*3]: "Ocean" includes ocean and coastal footprints, determined by GMI surface flag.

**Table 1: Collocation Statistics: total number of samples for each latitude band, season, surface condition (top: ocean and coastal; bottom: land), and scenario regime during March 2014 – October 2017. Note that total samples do not include collocations that are clear-sky or out of the boundaries of our definition of the four regimes.**





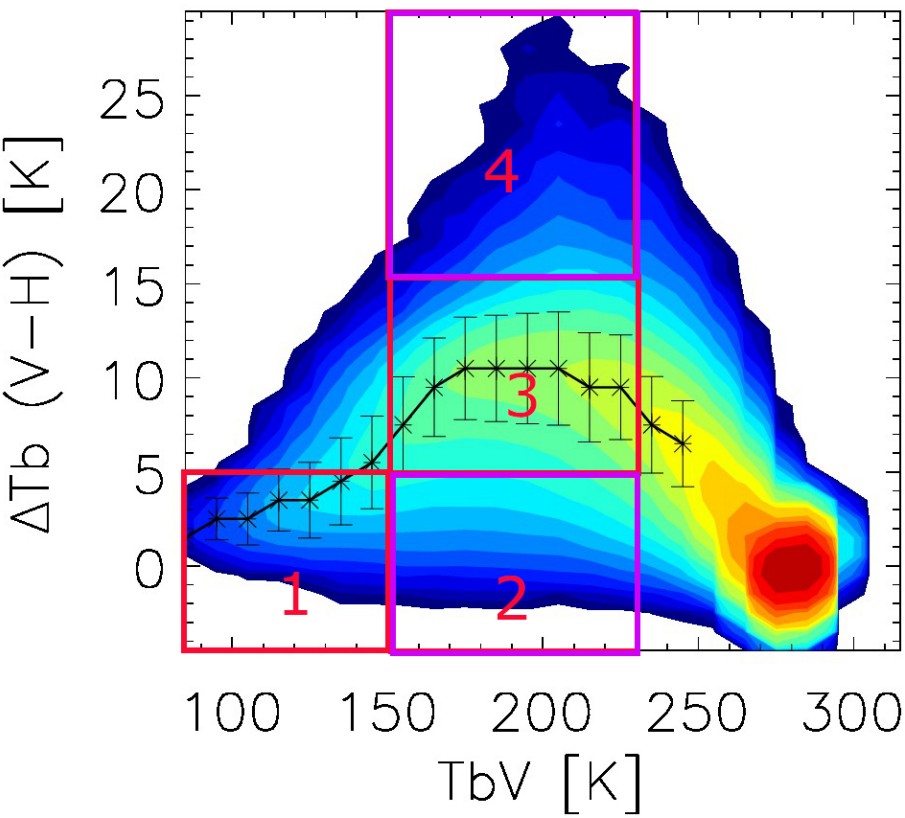


**Figure 1: The definition of PD regimes according to the TBv and PD values: (1) deep convective; (2) low-PD; (3) medium-PD; and (4) high-PD. See text for values regarding the regime definition. Regime (2) low-PD and (4) high-PD, enclosed by purple rectangles, are scenes we focus to study in this current work. Two-dimensional PDF contours are adapted from Fig. 3 of Gong and Wu [2017].**



**Figure 2: Contoured Frequency by Altitude Diagram (CFAD) from CloudSat (color shaded) and DPR-Ku (color contours) for the four regimes in Fig. 1 integrated from all tropical (30°$S$ − 30°$N$) collocated scenes. (a)-(d) corresponds to regime#1 - #4 respectively. Contour scale is linear and relative to its maximum power.**



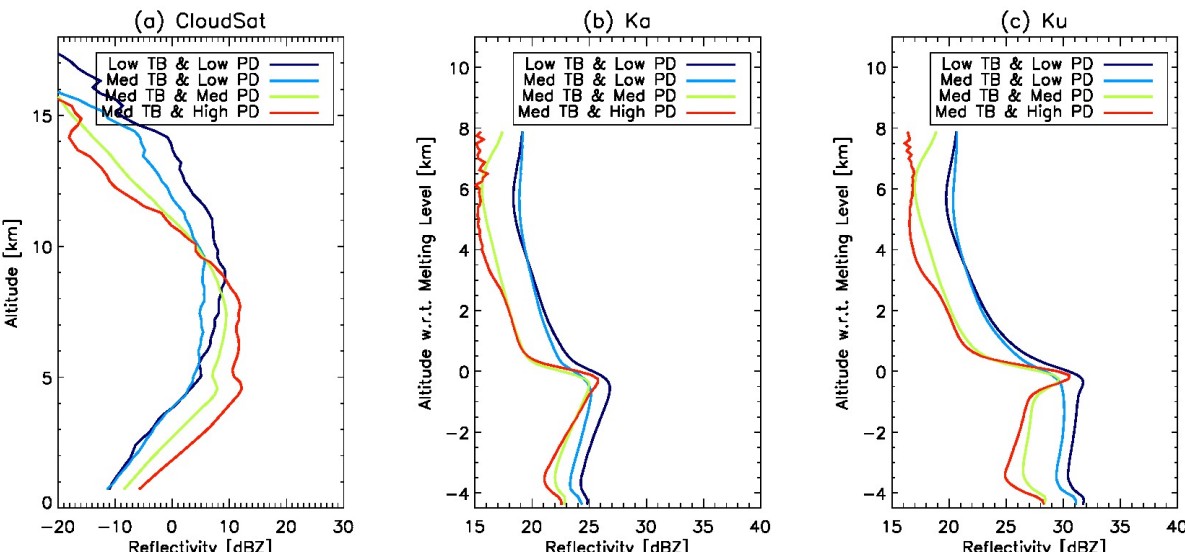

**Figure 3: The collapsed 1-D view of Fig. 2 after weighted averaging along the axis of absolute reflectivity (i.e., x-axis) for (a) CloudSat,**
**(b) DPR-Ka and (c) DPR-Ku. Note that the vertical axis for (a) is absolute altitude, while it's altitude with respect to melting level in (b) and (c). Red (light blue) line is for high-PD (low-PD) regime, respectively.**





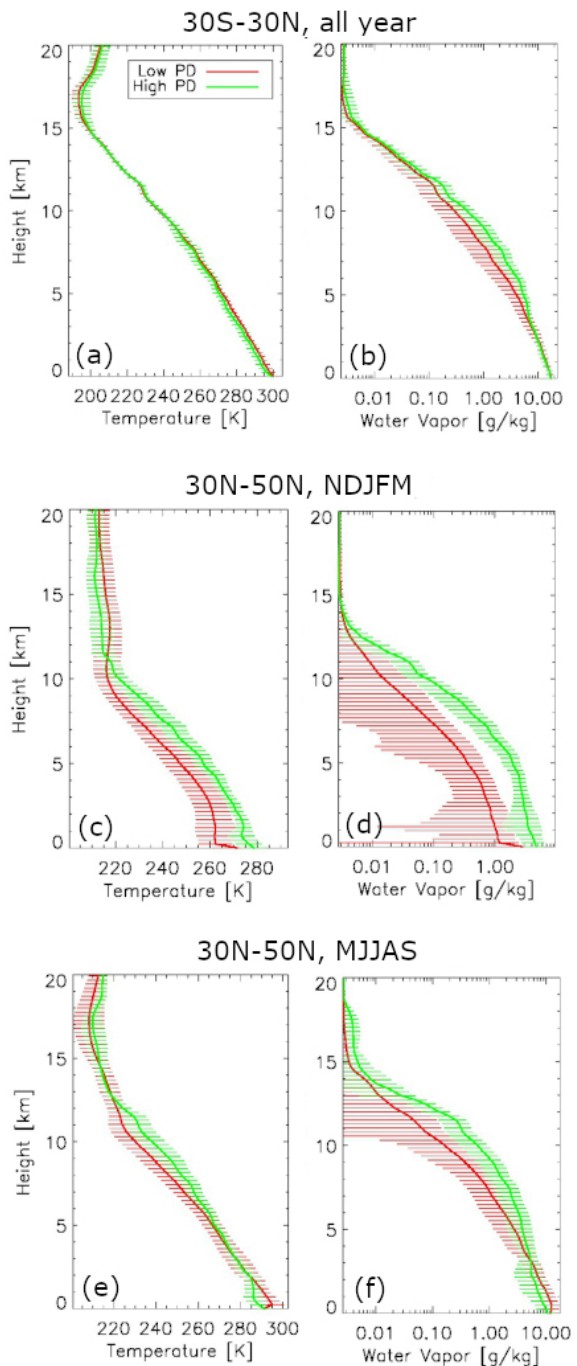

**Figure 4: Temperature (left) and water vapor (right) profiles for "high-PD" (green) and "low-PD" (red) scenarios, respectively, for tropics [$30°S - 30°N$] averaged over years (top), Northern Hemisphere extended winter (middle panels, $30°N - 50°N$, November-March), and Northern Hemisphere extended summer (bottom panels, May-September). Errorbars are also included in the same color for every other level.**





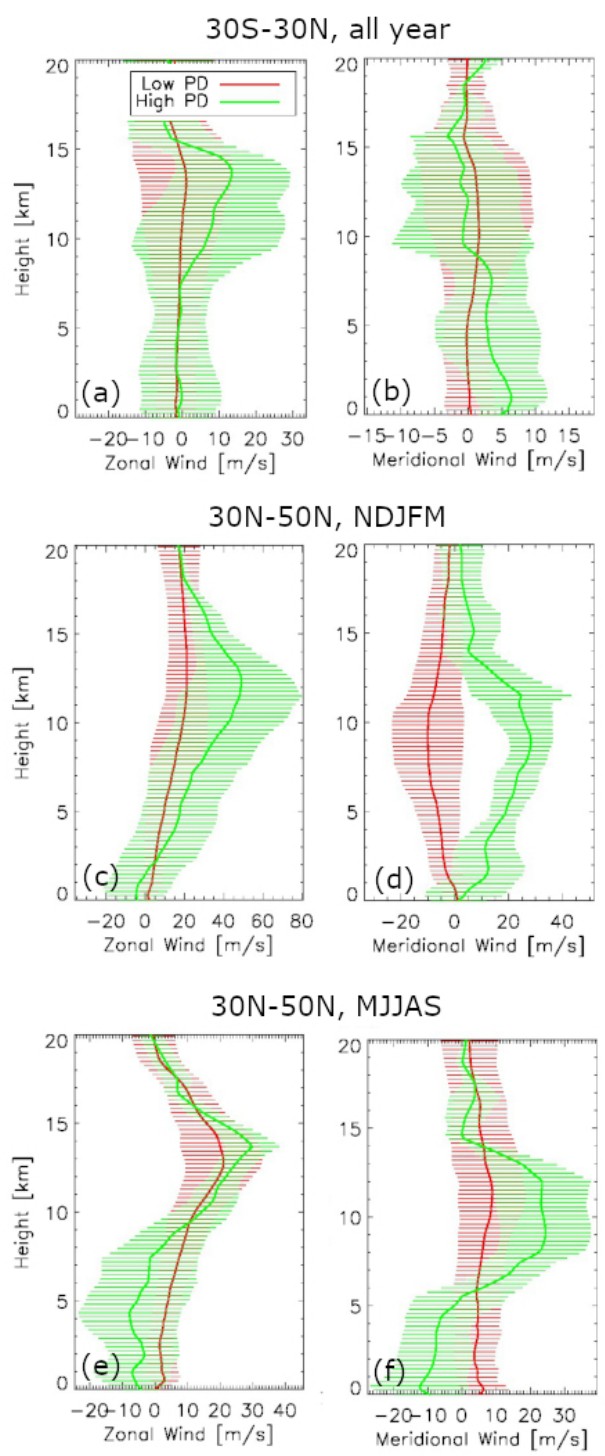

**Figure 5: Same with Fig. 4, except for zonal wind (left) and meridional wind (right).**




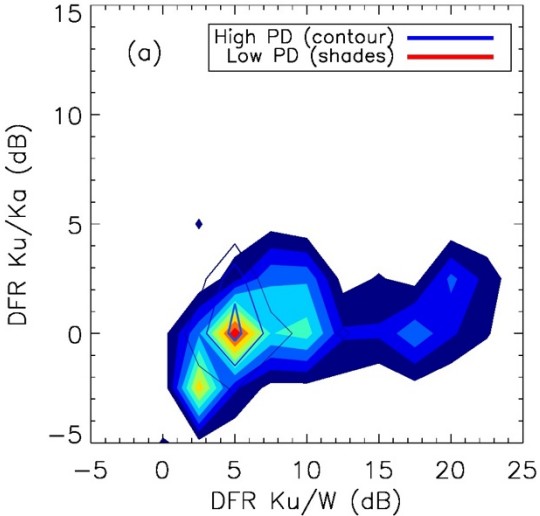 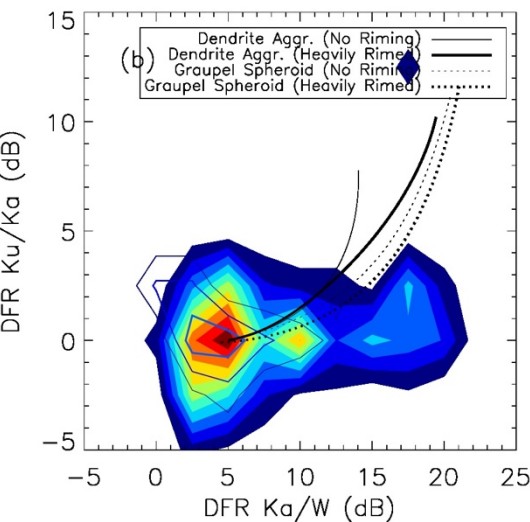

**Figure 6: 2-D distribution of triple frequency DFR diagram from high-PD (blue contours) and low-PD (color shaded) regimes composited from all collocated CloudSat and DPR observations: (a)** $DFR_{Ku/W}$ **versus** $DFR_{Ku/Ka}$**; (b)** $DFR_{Ka/W}$ **versus** $DFR_{Ku/Ka}$**. The reflectivity observations are taken from the altitude range of 5.5 km – 15 km to minimize impacts from melting layer or rain. Model simulated dendrite (solid) and graupel spheroid (dotted) behaviors are overlaid on (b) for unrimed (thin) and heavily rimed (thick) cases. See Leoinonen and Szyrmer [2015] for model and ice morphology details.**





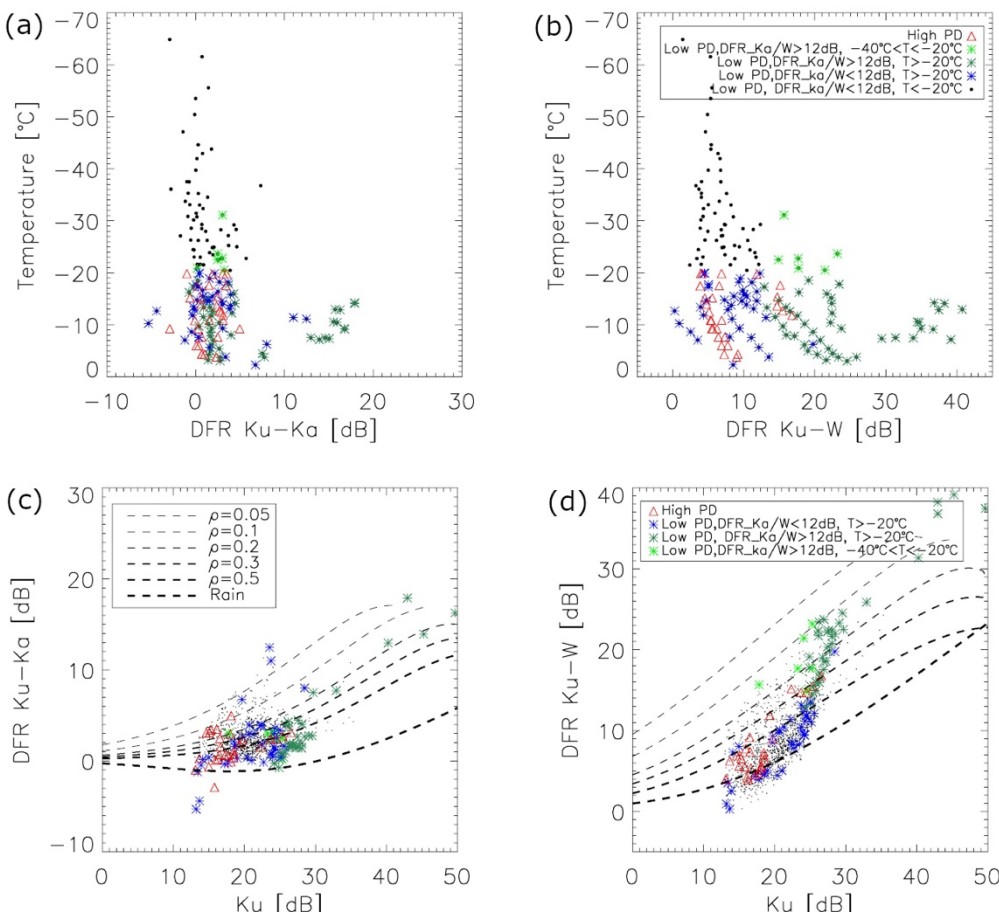


**Figure 7: Scatter plots of DFR relationship to temperature (top) and to reflectivity (bottom) for high-PD observations (red triangle), low-PD with cold temperature and small $DFR_{Ka/W}$ (black dots), low-PD with warm temperature and small $DFR_{Ka/W}$ (blue star), low-PD with cold temperature and large $DFR_{Ka/W}$ (green star) and low-PD with warm temperature and large $DFR_{Ka/W}$ (green-blue star). Theoretical calculation of density isolines are overlaid in the bottom panels to facilitate our understanding the density (and**
**habit) of each group of ice particles.**





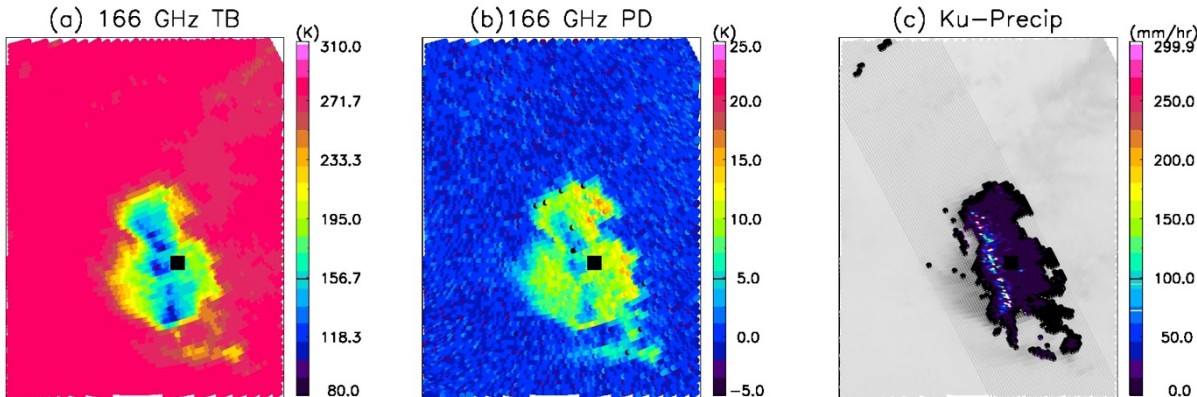

**Figure 8: A squall line case captured by GMI and DPR exhibits larger PD behind (to the right) the deep convective front than in the leading edge (to the left). (a) 166 GHz TB; (b) 166 GHz PD, and (c) KuPR retrieved surface precipitation rate (color) on DPR swath (grey) overlaid on top of GMI 166 GHz PD (light grey). The black rectangle is just a reference point for easy inter-panel comparison. This squall line case occurred on July 20, 2015 (orbit #007908) over Chad, Africa.**

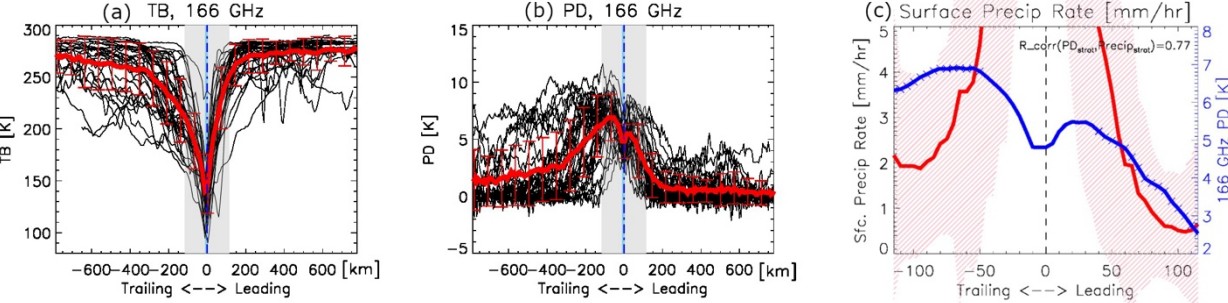

**Figure 9: 166 GHz TB (a) and PD (b) distribution across the squall line center (0 at horizontal axis) for all 46 squall line cases (black) and their mean (thick red) and standard deviation (thin red error bars). Trailing edge is to the left and leading edge is to the right. The squall line case details can be found in Table A1 in the Appendix. Light grey rectangles in (a) and (b) correspond to KuPR coverage. In (c), KuPR retrieved mean surface precipitation rate ($PR_{sfc}$) is shown as the thick red line across the squall line center with standard deviation shown in pink hatched areas. 166 GHz PD in the grey area in (b) is overlaid as the thick blue line in (c). Deep convective rainy footprints ($PR_{sfc} > 5 mm/h$) are excluded for plotting because our PD hypothesis only works in stratiform region. The Pearson rank correlation between $PR_{sfc}$ and PD is 0.77 (significance at 99.9% confidence level) after excluding the deep convective footprints.**





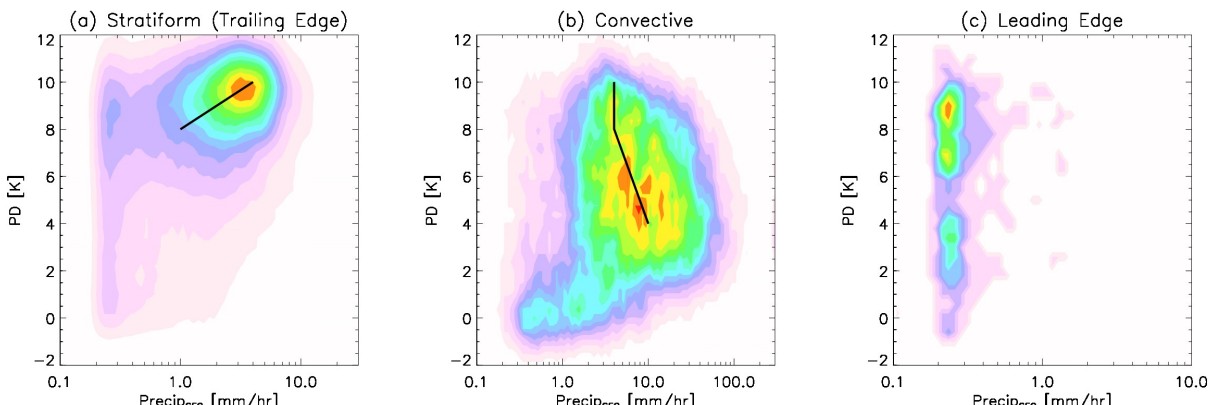

**Figure 10: Surface precipitation and 166 GHz PD distribution in the (a) stratiform, (b) convective and (c) leading edge of the squall lines. DPR stratiform/convective/other flags are employed to differentiate different precipitation types. Color scale is linear and relative to the peak value on its own PDF.**

## Appendix A: Complete list and geographic distribution of the selected squall line cases

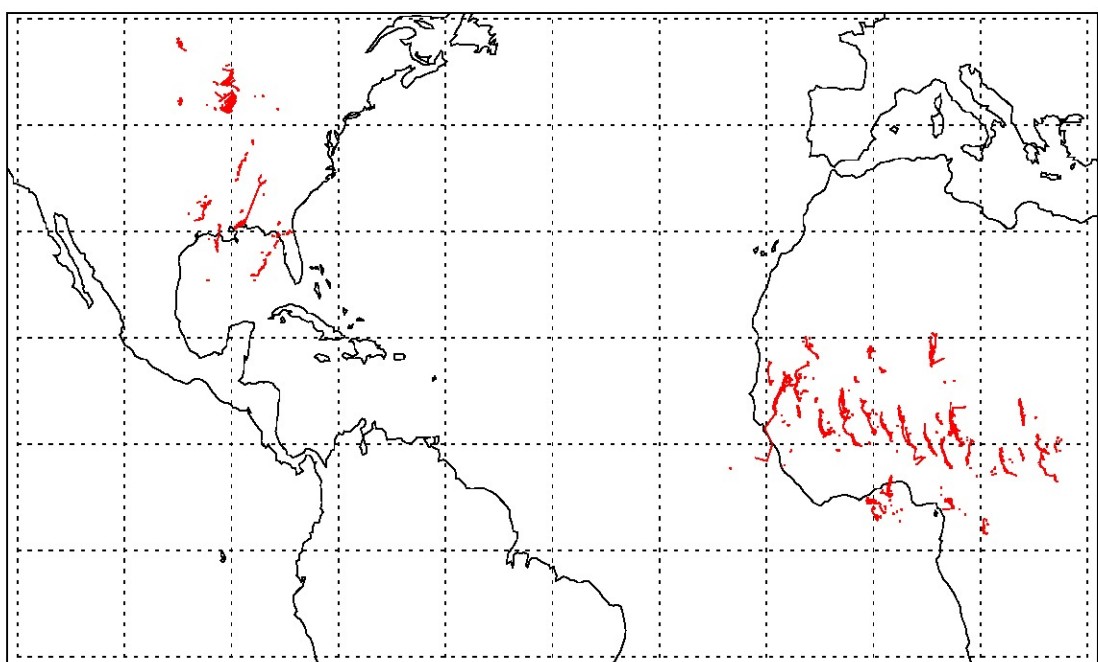

**Figure A1: The geographic distribution of the 46 squall line cases that are selected. Red dots are the location of identified deep convective center (i.e., maximum $PR_{sfc}$ while $PR_{sfc} > 25mm/h$ threshold) on each KuPR scan. Each scan swath is then aligned against this center location (setting as 0) to make the composites shown in Fig. 8 and Fig. A2.**



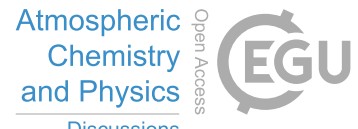

| Orbit Number | Year | Month | Day | Longitude Center [°E] | Latitude Center [°N] |
|---|---|---|---|---|---|
| 013718 | 2016 | 7 | 28 | 23.5 | 10.4 |
| 002737 | 2014 | 8 | 22 | 2.2 | 12.6 |
| 001476 | 2014 | 6 | 2 | 15.6 | 8.9 |
| 007202 | 2015 | 6 | 5 | -0.2 | 5.7 |
| 001461 | 2014 | 6 | 1 | 5.1 | 4.3 |
| 019220 | 2017 | 7 | 17 | 18.8 | 8.5 |
| 018584 | 2017 | 6 | 6 | -2.1 | 14.2 |
| 018891 | 2017 | 6 | 25 | 4.8 | 7 |
| 002169 | 2014 | 7 | 17 | -12.9 | 14.6 |
| 007779 | 2015 | 7 | 12 | 4.3 | 12.5 |
| 002245 | 2014 | 7 | 21 | 7.2 | 15.3 |
| 002098 | 2014 | 7 | 12 | 8.3 | 10.2 |
| 018691 | 2017 | 6 | 13 | 11.4 | 12.1 |
| 019451 | 2017 | 7 | 31 | 5.9 | 10.4 |
| 012965 | 2016 | 6 | 9 | 24.3 | 8.1 |
| 002483 | 2014 | 8 | 6 | -11.7 | 15.6 |
| 019021 | 2017 | 7 | 4 | -0.8 | 13.3 |
| 002744 | 2014 | 8 | 23 | -3.7 | 13.8 |
| 007908 | 2015 | 7 | 20 | 21.1 | 13 |
| 007640 | 2015 | 7 | 3 | 15.9 | 2.8 |
| 008225 | 2015 | 8 | 10 | -4.2 | 9.9 |
| 002753 | 2014 | 8 | 23 | -10.6 | 15.6 |
| 002107 | 2014 | 7 | 13 | -1.2 | 15 |



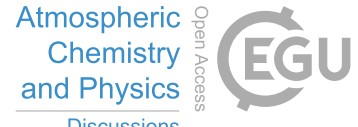

| | | | | | |
|---|---|---|---|---|---|
| 002029 | 2014 | 7 | 8 | 23.5 | 9.3 |
| 018737 | 2017 | 6 | 16 | 11.1 | 5 |
| 013934 | 2016 | 8 | 11 | 0 | 18.7 |
| 002114 | 2014 | 7 | 13 | -7 | 11.9 |
| 013097 | 2016 | 6 | 18 | 7.6 | 3.2 |
| 019267 | 2017 | 7 | 20 | -4 | 13.3 |
| 002675 | 2014 | 8 | 18 | 13.2 | 8.5 |
| 001768 | 2014 | 6 | 21 | 14.4 | 3.7 |
| 002660 | 2014 | 8 | 17 | 8.4 | 18.9 |
| 001791 | 2014 | 6 | 22 | 0.8 | 3.8 |
| 007847 | 2015 | 7 | 16 | 9.7 | 12.8 |
| 013734 | 2016 | 7 | 29 | 9.9 | 9 |
| 014211 | 2016 | 8 | 29 | -8.9 | 19.4 |
| 008308 | 2015 | 8 | 15 | 8.6 | 18.5 |
| 002851 | 2014 | 8 | 29 | 11.4 | 11.5 |
| 012959 | 2016 | 6 | 9 | 2.1 | 6 |
| 014225 | 2016 | 8 | 29 | 26 | 10.2 |
| 018861 | 2017 | 6 | 23 | -10.1 | 13.3 |
| 013419 | 2016 | 7 | 9 | 26.6 | 7.3 |
| 008309 | 2015 | 8 | 15 | -13.9 | 16.2 |
| 013795 | 2016 | 8 | 2 | 20.6 | 9.8 |
| 019245 | 2017 | 7 | 18 | -7.2 | 11.5 |
| 008414 | 2015 | 8 | 22 | -96.8 | 36.8 |



**Table A1: Details about the 46 squall line cases we selected. First column lists the GPM orbit number. 2nd-4th columns include year, month and day of the event, and last two columns are the longitude and latitude of the reference center (i.e., black rectangle in Fig. 8 for easy comparison, not necessarily the storm center).**

680

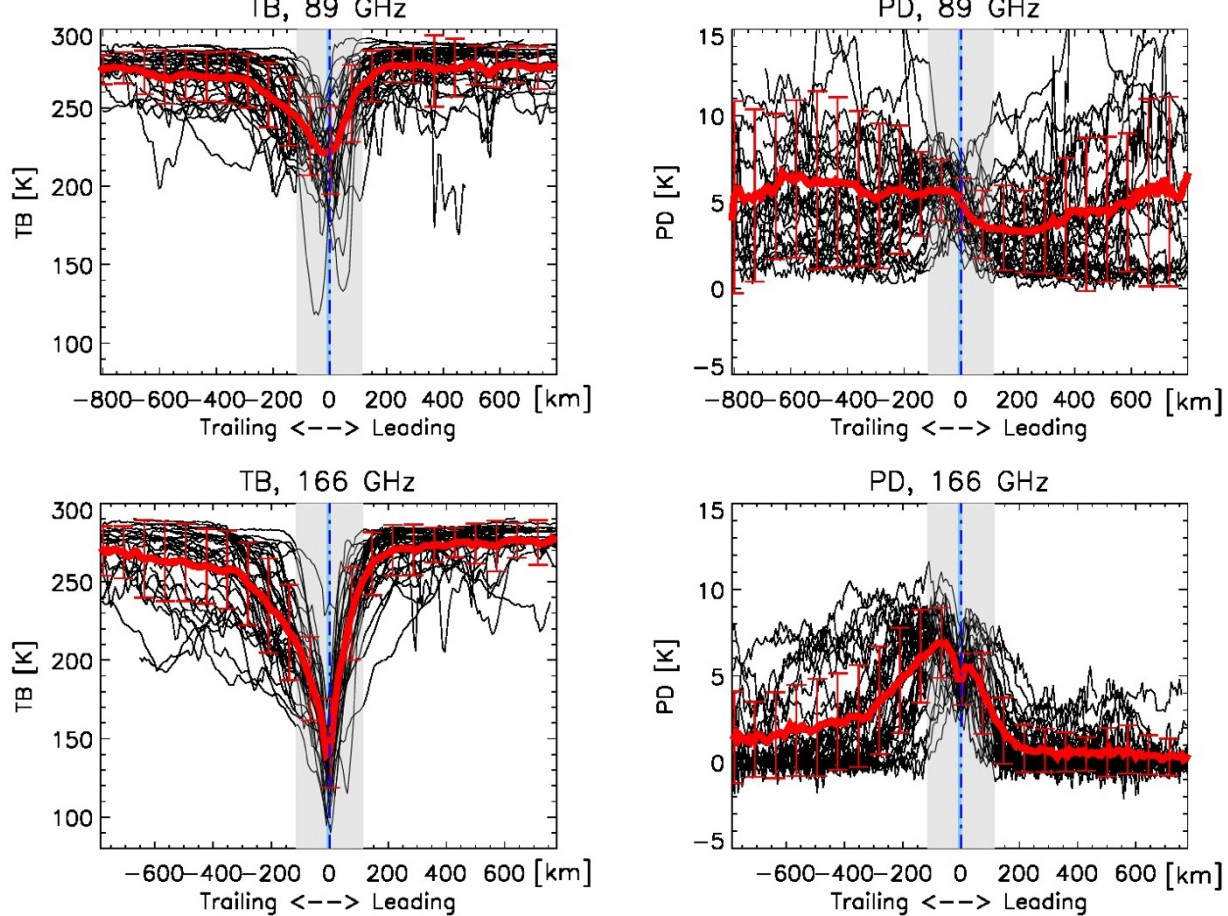

**Figure A2: The bottom panels are the same with Fig. 9a and 9b, and the top panels are similar except for 89 GHz. One can see that 89 GHz PD signal is not as clean as that at 166 GHz since 89 GHz's PD is also strongly impacted by surface wind and liquid cloud/raindrop.**