# Peer review of "Linkage among Ice Crystal Microphysics, Mesoscale Dynamics and Cloud and Precipitation Structures Revealed by Collocated Microwave Radiometer and Multi-frequency Radar Observations"

_Atmospheric Chemistry and Physics, 2020_

## Referee Comment (RC1) · Toshi Matsui (Referee) · 12 May 2020

General Comments)

This manuscript extends the analysis of polarization difference in high-frequency passive microwave brightness temperature along with triple-frequency radar backscattering. Overall, fairly well-written, and analysis and plots are generally reasonable (need a bit corrections). This paper definitely progressed the understanding of ice microphysics

in deep convection and stratiform precipitation. I really like the analysis of co-located sensors. I enjoyed reading it.

There are two specific comments and a number of technical corrections for improving the manuscript. Also, I'm not familiar with writing style of EGU, but you may improve the writing by omitting mathematical symbols and suppressing spoken language/expressions.

Specific Comments)

Life cycle vs. cloud structure: Line 218: can you weaken the word "conclude"? Actually, without tracking actual life cycle of deep convection, you cannot conclude your hypothesis. This could be explained by the location of cell (how close to the convective cores or geographic locations) in addition to the life cycle as seen in Fig 9. If you can, plot classification of categories 1-4 on Fig 8 case (like several case in horizontal map)? You may see all categories in one case of MCSs regardless of the life cycle of MCSs.

GPM DPR-CloudSat CPR mismatch: Line 353: You stated that you cannot trust Figure 7d due to imperfect match for microphysics analysis. In that case, you cannot also trust neither Ka/W nor Ku/W analysis in Figure 6? Can you justify why one analysis is trustable (Figure 6) and other (Figure 7d) is not.

Technical Corrections)

Line 38: "how much of that precipitation reaches the ground and where" -> "how much and whre precipitation reaches the ground"

Line 50: Add citations of triple-frequency radar retrievals.

Line 70: Remove "healthy"

Line 78: Computational cost is not the answer of using spherical assumption. It is just uncertain to derive size and orientation simultaneously.

Line 90: Please add citations of Olson et al. 2001 with several sentences in the introduction. This is probably most related original paper of using PD to discuss oriented non-spherical ice in the stratiform precipitation.

Line 114: "Microwave Imager" -> "GPM Microwave Imager"

Line 118: "250m" -> "500m" range resolution of interlaced DPR Ka band is 500m.

Line 121: "correction" -> "attenuation correction"

Line 148: Remove "(rain or ice)".

Line 157: Please write the resolution of ECMWF analysis.

Line 180: "convective scenes" -> "convective core".

Figure 2: Shade bar and values in contour lines are missing.

Line 207: "less and less" -> "lesser"

Line 208: "more and more thick" -> "thicker"

Line 215: "the melting layer" -> "the melting layer due to increased temperature"

Line 270: "presented evidences" is too strong. Suggest "Our analysis in Section 3 supports that….". Everything from remote sensing of microphysics is retrieval and guess. Without direct measurement of in-situ observation, you cannot conclude it.

Figure 6: Again, color shade bars and values in contours are missing.

Line 305: I'm not sure about these isolated sample. How significant it is. Can you still say the sample close to theoretical curve? It does not looks like.

Line 359: How do you roughly define "large" or "small" ice particle here?. What is the size ranges of ice particle

Line 439: "enjoy"?

Line 447-449: If you like to conclude this, you must show the plot using 89GHz (e.g.,

in Appendix). Otherwise, you cannot state it.

Line 451: "highest frequency dual-polarized radiance measurements" -> "highest frequency of dual-polarized microwave radiance measurements"

Line 475: You must also mention to use ground-based polarimetric radar for alternative approach, too.
* * *

---

## Referee Comment (RC2) · Anonymous Referee #1 · 25 May 2020

General Comments:

This study is focusing on the ice related microphysical processes and scattering properties, which are significantly difficult, especially from satellite platform, but play an important role in the Earth's radiation budget. I appreciate that this study uses passive microwave measurements as the primary dataset, combined with space-borne radar observations to gain insight into the vertical structures of the ice properties. I com-

mend the authors for an interesting manuscript and an extensive work. I really like the comprehensive analysis in sections 4 and 5, especially when relating PD variations to different cloud life stages. It is a very interesting angle of looking at the deep convective cloud properties. However, I do have some major concerns mainly related to Section 3.

Major concerns:

1. It is interesting to investigate which condition promotes higher PD, but when authors discuss the background atmosphere differences between High-PD and low-PD cases, the large-scale environment data that used in the analysis are actually from in-cloud or partially in-cloud pixels. This large-scale thermodynamic and kinematic fields have already modified by the convective systems. It is not representative of the environmental or thermodynamic conditions that the convective systems initiate and develop. The pre-storm thermodynamic profiles that prior to the convection should be used.

2. The differences in large-scale conditions between High-PD and low-PD are found to be fairly small in the Tropics. I am wondering how much of this just from the land-ocean contrast or seasonal variabilities (wet vs. dry). It's worth further development.

3. One thing I think necessary is to provide more context/details for certain aspects of the study, such as the radiative transfer simulation setup and assumptions, and necessary references for certain sentences.

4. The colorbars are missing for almost all the shading plots.

Minor comments:

1. L17. Specify 'high-frequency'.

2. L77. "while some of the recent products . . .", here needs references.

3. L90-91. Give full names to TMI and MADRAS.

4. L110. This section lacks references for the datasets and the instruments overall.

5. L129. In the paragraph, authors use "PD-TB", it is actually "PD-TBV". Please keep it consistent throughout the whole manuscript.

6. L137. "the PD-TB relationship is largely latitude-independent..." needs references for this sentence. Even though the mean tends to be similar, but the standard deviation may be different between tropical and high latitude events, which could add potential uncertainties to the regime definition. It would be nice to see the results for midlatitude and high latitude, e.g., Figure 2.

7. L139-L142 are confusing. Do you mean congestus in general lacks stratiform clouds? I do not think the reason for including shallow clouds should be the difference in area fraction of convective core and widespread stratiform. Please re-phrase it.

8. L162. "This dataset has been used by many other researcher..." Please provide references.

9. L178-180 Please provide references.

10. L208. Authors should make it clear to readers why regime 1 is defined as "deep convection".

11. L239, L244, see major comment #2.

12. L249. Do these differences pass significant tests?

13. L283. RTM needs to be defined.

14. Figure 6. color bars are needed. Is this for the whole data or just tropical cases? The legend on Figure 6(a) is wrong.

15. L360. ICI needs to be defined.

16. Figure 9. The SD is very hard to see. color bar is missing.

---

## Referee Comment (RC3) · Anonymous Referee #2 · 25 May 2020

In this article, Gong et al. delve into a unique GMI measurement, the difference between Polarimetric Difference (PD) between radiances at 166 GHZ, that could be used to retrieve novel information for precipitation retrievals and satellite-based meteorological research. I believe the science of the study is incredibly valuable. PD represents a unique source of information in frozen clouds that can only be retrieved from the GMI, but interpreting radiance information on frozen precipitation has been a challenge due to the multiple explanations that could account for a similar PD signal. The authors

provide a clever combination of globally averaged observations between two satellites, simulation results, and a set of specific storm observations to winnow down the physical interpretation of PD observations. I look forward to following future developments in this research.

Unfortunately, I believe the writing quality for this article falls below the level necessary for publication. Grammatical errors are frequent (e.g. "collocation cases... are averaged separately considering they locate in different weather regimes at L228), articles/prepositions are often missing or used inappropriately, and word choices are often inappropriate, both in the sense of conventional English (e.g., "the biggest blob of power" at L304) and in terms of formal writing (e.g., "more and more" throughout). Occasionally, it was enough of an issue that I found it difficult to understand the author's message. Please make sure that this paper is proofread more carefully before the next submission, with specific attention to formal word choices.

Specific comments are below:

L40: It would be nice to have more detail here. I have never heard of Cloud Radiative Effect before, and I would like to know which microphysical properties you are referring to.

L17/87: PD is defined in two different ways (Polarimetric radiance Difference, Polarimetric Difference). I would suggest unifying these.

L163: Has the collocated GMI, DPR, and CPR dataset been used by "many" other researchers for published articles? Yin et al. (2017) was the only one I was familiar with before reading this paper.

L193: ". Compared with "high-PD" scenario, the "low-PD" one apparently has more high clouds that are thinner than those from deep convective scenes as the reflectivity magnitudes are smaller". Unsure of what you mean by "more high clouds that are thinner". According to the CloudSat CFAD, the distribution of reflectivities for clouds

above 13 km appear to extend larger at low PD than high PD, so I feel like that would make the high clouds at low-PD thicker?

L203: I don't understand how a mean could be "PDF-weighted". Consider rewording?

L203-L218: I don't agree with the interpretations of reflectivity used throughout this paragraph, and I would prefer if the authors would be more objective. Reflectivity is used to describe the "thickness" of clouds and the "presence" of clouds in the same sentence, even though these two traits are not directly comparable. Later it is used to describe the number of precipitation sized particles (less/more), even though reflectivity can also be indicative of precipitation particle size. I would prefer if the figures are discussed more literally in this section, just explaining which PD scenarios have larger average reflectivity at different altitudes, and then save the interpretation of reflectivities for the following paragraph.

Figure 3: I do not see any novel information provided by Ka band reflectivity in Figure 3b, so I think it could be removed.

On a similar note, Ka band reflectivity can be used to construct a Ku/Ka DWR. DWR can be used to provide information on ice particle size without being influenced by particle concentration, unlike the Z measurements used throughout the study. Comparison between DWR and Ku may also provide information on ice particle concentration. Considering the frequent discussion on ice particle aggregation, I think it could be valuable and relatively straightforward, to make figures similar to Figure 2 and 3 (and potentially Figure 8) with DPR-measured DWR. Keep in mind that DWR may also be influenced by liquid water attenuation, so a DWR profile should be interpreted with caution. This is more of a suggestion than a perceived requirement for publication, however.

Figure 6: I am having difficulty understanding this figure well enough to determine whether I agree with your interpretation. The combination of filled and unfilled contours worked in Figure 2, but only because the zones of reflectivity were mostly separate. I think this figure would work better if both high PD and low PD were line contour plots of

different colors, and regardless of whether that suggestion works, I would appreciate if this figure could be replotted to be more decipherable.

---

## Referee Comment (RC4) · Anonymous Referee #4 · 30 May 2020

General comments:

I would like to begin by commending the authors for their extensive and innovative work carried out in this study. I'm grateful for the opportunity to review it.

The combined usage of active and passive measurements is clever. Passive measurements polarisation difference provides information on the orientation of the ice particles and the active sensors provides information on the vertical structure of the clouds. This

allows for the disentanglement of various microphysical processes in the convective and stratiform cloud systems. The paper demonstrates the value of passive polarization sensors and is scientifically very important. I don't have any major objections to the scientific content, apart from the comments given below.

The structure and analysis of the paper is clear. Figures are generally informative and relevant; I found all of them interesting. Importantly, the limitations of the used data are clearly presented.

Unfortunately, the paper is lacking significantly in terms of grammar and phrasing. Some sentences are difficult to read and understand. Examples are found under grammar/technical comments below. Careful proofreading is required before publication.

Major comments:

1. While theoretical calculations are not the focus of the paper, only meant to augment the analysis, I think they could be described in more detail. For instance, what particle size distribution was assumed in the triple frequency calculations? More information can be given on the habits and the assumptions made for the riming model. This should not need more than two paragraphs I believe.

2. There is a lack of discussion regarding the sample size of the used data. Naturally, the amount of collocated GPM and CloudSat measurements of relevant cloud types are limited outside the polar regions. However, there is a lack discussion on how this could affect the analysis. There are only 62 high PD samples in total (table 1), meaning that features visible for the low PD data might not be captured for high PD data (in figure 6 for instance). How does this affect the credibility of the conclusions made in this study?

Specific comments:

1. L132: Could you please provide some motivation or background for the regime limits, especially the 150 K limit for deep convection.

2. L198: "Because the KuPR reflectivity does not saturate with particle size as rapidly

**ACPD**
as Cloudsat, we can also infer large ice particles high in the atmosphere in the deep convective and low PD cases...". Do you mean that the limited sensitivity of KuPR to smaller ice hydrometeors indicates that high altitude KuPR reflectivities are due to large particles? Perhaps this could be phrased better.

3. L209: Could you perhaps use some other word than regime? Regime is already used (high-PD, deep convective, etc), so this caused some confusion for me. Perhaps "mode" is better?

4. L215: Please rephrase the sentence. Suggestion: "The latter scenario indicates the late stage of a convection life cycle, where the convective cell disappears and a stable stratiform layer forms to dominates the whole column."

5. L262: What types of observations are referred to here? Reference?

6. L283: Should not the unit of the reflectivity ratio be unitless (or dBZ)? Confusing.

7. L294: I think it is prudent to also refer to Toyoshima (2015), for the DPR thresholds (Yin et al. (2017) added the CPR threshold).

8. L294: Forgive me if I've missed this, but I can't find any description on the graupel spheroid in Leinonen and Szyrmer (2015). I suggest you provide at least a short description of both the habits and also on how the riming is modelled. I also find it contradictory to talk about a graupel spheroid under no-riming conditions, since graupel implies growth by riming.

9. L296: Why is repositioning of the theoretical curves necessary? To augment the analysis? Please explain.

10. L339: How is density defined in this context?

11. L344: Sentence is very long and difficult to read.

12. L354: "evidences" -> "evidence". However, given the limitations (including sample size), I wonder if it would not be more prudent to use "indications" instead. I don't think
this would take away from the novelty and importance of this work.

13. Section 6: I'm of the opinion that this section could better summarize the key conclusions and future work, in a more concise way. As it is now, it is rather a summary of what was presented and discussed in the paper. What are the take-home messages of this article?

14. Figure 3: Could this figure include standard deviations (in dashed lines for instance)?

15. Figure 5: I found it surprising that high PD signals are prevalent even in the presence of high wind speeds. Shouldn't high wind speeds promote random orientation? Just a comment.

Grammar/technical comments:

1. L14: "ambient environment" -> "the ambient environment"

2. L15: "...impact up to the future climate projection and down to the details of the surface precipitation". Needs rephrasing. Suggestion: "...have on impact on climate projections as well as on the details of surface precipitation".

3. L18: Remove "are".

4. L43: "cloud" -> "clouds"

5. L55: "tornado" -> "tornados"

6. L72: Sentence needs rephrasing. Suggestion: "With swath widths typically over 1000 km and footprint sizes of 7-15 km, their combined usage can readily generate ice hydrometer producton temporal and spatial scales that suits the needs of both weather and climate studies."

7. L77: Problematic sentence. Suggestion: "While some of the recent products have advanced from using spherical ice models to more realistic habits, random orientation

ACPD
is still nearly always assumed, as it reduces computational complexity and decreases the degree of freedom for the otherwise severely under-constrained inversion problem."

8. L81: "putting" -> "put"

9. L162: Sentence difficult to interpret. Should it be: "For example, Yin et al. [2017] used collocated CPR-DPR reflectivity profiles from this dataset to study the discrepancies found in triple frequency radar signatures and inferred different microphysical processes in convective and stratiform regimes."?

10. L245: Sentence difficult to difficult. Please rephrase.

11. L253: It is difficult to understand what "which" refers to in the previous clause. Please rephrase for more clarity.

12. L297: "property"  $\rightarrow$  "properties"

13. L303: Try use a more formal word than "blob". Perhaps "accumulation"?

14. L356: "signal" -> "signals"

15. L365: Problematic sentence. Suggestion: "It is well established that anvil clouds are likely associated with low-PD signals, while high-PD signals are instead linked to stratiform layers."

16. L400: I think fig. 10 is introduced a bit suddenly here.

17. L415: Sentence is difficult to understand. Please rephrase.

References:

Toyoshima K, Masunaga H, Furuzawa FA. Early Evaluation of Ku- and Ka-Band Sensitivities for the Global Precipitation Measurement (GPM) Dual-Frequency Precipitation Radar (DPR). SOLA 2015;11:14–17. Interactive comment

2020.

---

## Author Response (AR1)

Response to Reviewers' comments:

RC1:

We highly appreciate Dr. Matsui's in-depth review and constructive suggestions. The English/grammar errors pointed by this and other reviewers have been corrected. The revised draft has been carefully proofread by the 4th author, who is a native speaker. The missing colorbars have now been added back. Below are point-by-point responses (questions in black, and responses in blue).

Life cycle vs. cloud structure: Line 218: can you weaken the word "conclude"? Actually, without tracking actual life cycle of deep convection, you cannot conclude your hypothesis. This could be explained by the location of cell (how close to the convective cores or geographic locations) in addition to the life cycle as seen in Fig 9. If you can, plot classification of categories 1-4 on Fig 8 case (like several case in horizontal map)? You may see all categories in one case of MCSs regardless of the life cycle of MCSs.

We totally agree with the reviewer that sometimes the tone was too strong, and indeed we cannot make a concrete conclusion without observing the entire life cycle of many convective systems, which is only possible on a geostationary platform from spaceborne point of view. Here at Line 218, we've replaced "conclude" with "summarize". We also add one more sentence reads as "Without actually tracking the entire life cycle of convective system(s), these arguments are just speculations. We will show some supportive evidences in Section 5 using an ensemble of squall lines. Furthermore, at this point, we cannot yet determine whether…".

GPM DPR-CloudSat CPR mismatch: Line 353: You stated that you cannot trust Figure 7d due to imperfect match for microphysics analysis. In that case, you cannot also trust neither Ka/W nor Ku/W analysis in Figure 6? Can you justify why one analysis is trustable (Figure 6) and other (Figure 7d) is not.

Since Ku/Ka is perfectly matched, but Ku/W and Ka/W are not, we can in principle compare DFR_Ku/Ka directly with theoretical calculations as long as we believe the theoretical calculation is correct, but only compare DFR_Ku/W and DFR_Ka/W qualitatively with theoretical calculations. Comparing Fig. 6 with Fig. 7, the main features in Fig. 6a and 6b agree with each other, so although both of them have caveats, we choose to keep them as they are telling consistent stories. For Fig. 7, since some features in the left do not match those in the right panels, we choose to interpret results to the left because they are from perfect matches, while leaving the right panels as "less trustworthy". The tone has been re-tuned to clarify our points.

Technical Corrections

Line 38: "how much of that precipitation reaches the ground and where" -> "how much and where precipitation reaches the ground"

Corrected. Thanks.

Line 50: Add citations of triple-frequency radar retrievals.

Chase et al. [2018, GRL] has been added, which demonstrates the capability of revealing ample frozen particle characteristics using the triple-frequency radar retrievals. In addition, the 5 citations included in the line above are also related to triple-frequency radar retrievals, so they are not replicated again here.

Line 70: Remove "healthy"

Removed. Thanks.

Line 78: Computational cost is not the answer of using spherical assumption. It is just uncertain to derive size and orientation simultaneously.

Thanks for the suggestion. Now the sentence has been altered as "random orientation is still nearly always assumed to avoid the complexity of deriving size and orientation simultaneously, as well as to avoid solving equations for 4 Stokes parameters simultaneously".

Line 90: Please add citations of Olson et al. 2001 with several sentences in the intro-duction. This is probably most related original paper of using PD to discuss oriented non-spherical ice in the stratiform precipitation.

A sentence has been added to Line 93-94, which reads as "In particular, Olson et al. [2011] used TMI 89 GHz PD as one of the several parameters for stratiform/convective precipitation classification." This paper is also discussed later (Section 5, around Line 451-453).

Line 114: "Microwave Imager" -> "GPM Microwave Imager"

Added. Thanks.

Line 118: "250m" -> "500m" range resolution of interlaced DPR Ka band is 500m.

I checked the ATBD of DPR Level 2 retrieval products, and I believe the vertical resolution is 250m.

Line 121: "correction" -> "attenuation correction"

Added. Thanks.

Line 148: Remove "(rain or ice)".

Removed. Thanks.

Line 157: Please write the resolution of ECMWF analysis.

After checking the ATBD of ECMWF-AUX dataset, we realized they used 0.5-deg 3-hourly forecast data for interpolation to CloudSat orbit time/location. The text has been updated and the new citation "Cronk and Partain, 2017" has been updated to replace the P_04 ATBD.

Line 180: "convective scenes" -> "convective core".

Change made. Thanks.

Figure 2: Shade bar and values in contour lines are missing.

Colorbar added now. Thank you.

Line 207: "less and less" -> "lesser"

Corrected. Thanks.

Line 208: "more and more thick" -> "thicker"

Corrected. Thanks.

Line 215: "the melting layer" -> "the melting layer due to increased temperature"

Added. Thanks.

Line 270: "presented evidences" is too strong. Suggest "Our analysis in Section 3 supports that. . ..". Everything from remote sensing of microphysics is retrieval and guess. Without direct measurement of in-situ observation, you cannot conclude it.

Thank you. Change made and toned down.

Figure 6: Again, color shade bars and values in contours are missing.

Colorbar added now. Thank you.

Line 305: I'm not sure about these isolated sample. How significant it is. Can you still say the sample close to theoretical curve? It does not look like.

Out of the 704 "low-PD" samples between 5.5 – 15 km height range, 128 of them fall into the range to make the color shaded contours in Fig. 6a and 6b. While 89 of them make the low DFR blob, the rest 39 samples make the large DFR_ka/w and small DFR_ku/ka blob to the right, so they are not from isolated samples. The difficulty we face here is that the theoretical curve can only account for a small portion of variability that we observed from collocated CloudSat-DPR, but the latter group of 39 samples are closer to the heavily rimed particle calculation.

Line 359: How do you roughly define "large" or "small" ice particle here? What is the size ranges of ice particle?

We admit that "large" and "small" are relative to each other, and we don't have a clear size range for each of them. Here "small" means cloud ice particles that have negligible fall speed, while "large" particles are precipitation-sized ones that have a non-trivial fall speed. They roughly separate at ~ 100 um mass weighted effective radius.

Line 439: "enjoy"?

We do feel it is appropriate to use a metaphor here, so no change is made.

Line 447-449: If you like to conclude this, you must show the plot using 89GHz (e.g.,

in Appendix). Otherwise, you cannot state it.

We included "(Fig. A2 in the Appendix)" in the parentheses now. Thanks.

Line 451: "highest frequency dual-polarized radiance measurements" -> "highest frequency of dual-polarized microwave radiance measurements"

Suggestion incorporated. Thanks.

Line 475: You must also mention to use ground-based polarimetric radar for alternative approach, too.

Sorry that we are not quite sure how ground-based polarimetric radar can help delineate the intertwined size/orientation/size/riming microphysical properties. Would you please suggest some references? Based on my standing, if it's single frequency polarimetric radar, it can tell more information about density and fall speed, but not very helpful on retrieving other parameters.

RC 2:

We highly appreciate Reviewer #2's in-depth review and constructive suggestions. The manuscript
Below are point-by-point responses (questions in black, and responses in blue).

Major concerns:

1. It is interesting to investigate which condition promotes higher PD, but when authors discuss the background atmosphere differences between High-PD and low-PD cases, the large-scale environment data that used in the analysis are actually from in-cloud or partially in-cloud pixels. This large-scale thermodynamic and kinematic fields have already modified by the convective systems. It is not representative of the environmental or thermodynamic conditions that the convective systems initiate and develop. The pre-storm thermodynamic profiles that prior to the convection should be used.

We used ECMWF-AUX dataset generated by CloudSat team, which is basically interpolating ECMWF 3-hourly 0.5 X 0.5 degree forecast to CloudSat footprint [Cronk and Partain, 2017 in references]. At this spatial resolution, deep convective core is largely unresolved but parameterized, although the mesoscale system is resolved in certain sense. Therefore, we feel it's appropriate to consider ECMWF-AUX data as the "background" atmosphere, and admittedly this is not a strict definition of "background" as it contains in-cloud and ambient circulation information (first paragraph in Section 3.2). Interestingly, one of this paper's coauthors had a completely opposite argument to this reviewer. She believed from mesoscale modeling point of view that the dynamic and thermodynamic conditions from ECMWF-AUX cannot represent any in-cloud circumstances, unless we were to use a cloud-resolving model simulation. Since Fig. 4 and 5 and related context are based on statistics of many samples, we believe it is representative of the ambient circulation difference.

2. The differences in large-scale conditions between High-PD and low-PD are found to be fairly small in the Tropics. I am wondering how much of this just from the land-ocean contrast or seasonal variabilities (wet vs. dry). It's worth further development.

This is an excellent point. We never thought about that direction. We've tested on separating the tropical samples to land vs. no-land (ocean+coastal), but we found no distinct differences. This is more or less expected to see because abundant water vapor in the tropics really smear out ocean-land contrast at 166 GHz when it's cloudy-sky. We didn't test the seasonal variability because sometimes we have 0 samples for a given scenario (e.g., dry season Indian monsoon plus American monsoon area contains 0 samples of "high-PD" scene, which doesn't mean there's no high-PD pixels, but just mean there's no collocated CloudSat-DPR observations that happen to have a "high-PD" GMI reading). We cannot construct meaningful statistics based on too few samples.

3. One thing I think necessary is to provide more context/details for certain aspects of the study, such as the radiative transfer simulation setup and assumptions, and necessary references for certain sentences.

The triple-frequency DFR simulations share the same set-up and scattering database with Leinonen and Szyrmer [2015], which was included at the end of 3rd paragraph. Simulated density isolines are replicated from [Liao and Meneghini, 2011], which were not explicitly explained but now has been included. Since the focus of this work is not to develop nor validate RTMs, these two sets of simulations are rather employed to facilitate qualitative interpretation of the observed features.

4. The colorbars are missing for almost all the shading plots.

Since the PDFs are always calculated to reference to the maximum value and the colorbar is linear, they are neglected. We realize this should not be omitted in a scientific paper. Now they are all added back. Thanks for the suggestion.

Minor comments:
1. L17. Specify 'high-frequency'.

High-frequency is channel frequency > 150 GHz. This has been added. Thanks.

2. L77. "while some of the recent products . . .", here needs references.

An example is given in the parentheses now: "…realistic habit [e.g., MODIS collection 6 assumed a bulk column-aggregate globally for its ice cloud properties retrieval, Platnick et al., 2017]."

3. L90-91. Give full names to TMI and MADRAS.

The acronyms have been spelled out now. Thanks.

4. L110. This section lacks references for the datasets and the instruments overall.

Two references (Skofronick-Jackson et al., 2018; 2019) and GPM website are now added. Thanks.

5. L129. In the paragraph, authors use "PD-TB", it is actually "PD-TBV". Please keep it consistent throughout the whole manuscript.

Thank you for notifying this mistake. We've corrected three places that PD-TB appears.

6. L137. "the PD-TB relationship is largely latitude-independent. . ." needs references for this sentence. Even though the mean tends to be similar, but the standard deviation may be different between tropical and high latitude events, which could add potential uncertainties to the regime definition.

The latitude-independence is reported in Fig. 4 of Gong and Wu [2017]. See the figure below for an example of 166 GHz global ocean statistics (left panel), and 45N land PDF (right panel). We can see that the 45N PDF looks not much different from that of the tropics. At higher latitude, the range of TB is smaller so we can only construct the right portion of the upside down bell-curve (dark red line in the left panel for example), but the curvature, the PD peak and where the peak occurs remain largely the same.

[Figure]

7. L139-L142 are confusing. Do you mean congestus in general lacks stratiform clouds? I do not think the reason for including shallow clouds should be the difference in area fraction of convective core and widespread stratiform. Please re-phrase it.

What we mean is that GMI 166 GHz only cannot differentiate congestus and anvil cloud/stratiform deck. Below is an example given in Zeng et al. (2018, https://gpm.nasa.gov/sites/default/files/meeting_files/PMM%20Science%20Team%20Meeting%202018/Posters/%23103_Zeng.pdf), clouds in circle B are congetus, while clouds in circle A are anvil. Their types are determined by collocated CloudSat vertical scan (black line). Congetus show similar TB and PD ranges with anvil and stratiform clouds, so they cannot be differentiated out. However, readers can also see the areas of congestus covered are much smaller than anvil clouds, so we don't think including congestus will significantly bias our statistics. We have changed the last sentence to "As congestus area is

much smaller than anvil/stratiform precipitation areas [e.g., Zeng et al., 2018], the immixture of shallow convection structures should have negligible impact on the general statistics."

[Figure]

8. L162. "This dataset has been used by many other researcher. . ." Please provide references.

An example (Yin et al., 2017) has been given in the following sentences. Our previous works, including Gong et al. (2017) and Zeng et al. (2019) also employed this dataset. These two citations have been included. Thanks.

9. L178-180 Please provide references.

A reference of Kirstetter et al. [2014] has been added, where the authors have found consistently larger beam-filling effect due to subpixel inhomogeneity for convective rainfall versus stratiform rainfall using TRMM 2A25 product.

10. L208. Authors should make it clear to readers why regime 1 is defined as "deep convection".

We agree with the reviewer that there is a counter-intuitive logic here in our definition. Usually people define "deep convection" based on the maximum radar reflectivity passing a certain threshold, which is more or less arbitrary as well, and the definition between CloudSat and DPR are not consistent because they work at different frequencies. Our definition is purely based on a 166 GHz TB threshold (TB < 150 K). This is of course arbitrary too.

If you revisit left panels in Fig.A2, you can see 150K corresponds to the very deepest depression of 166 GHz TB, which is the center of the deep convective line. Note that to make Fig. A2, we don't set up any threshold but simply assign the coldest TB at each scan as the center of convective core. At 89 GHz this deep convective core ensemble is roughly < 225K. This value has been used previously in literatures to identify deep convections from TRMM TMI 85 GHz (e.g., Spencer et al., 1989; Nesbitt et al., 2000). So our 166 GHz threshold is consistent with 85 GHz threshold that studies used before. These two citations are now included to support our regime definition.

11. L239, L244, see major comment #2.

Please see our reply under major comment #2.

12. L249. Do these differences pass significant tests?

Yes, they do pass student-t test at 95% significance level.

13. L283. RTM needs to be defined.

The acronym has been spelled out now. Thank you.

14. Figure 6. color bars are needed. Is this for the whole data or just tropical cases? The legend on Figure 6(a) is wrong.

It's from the whole data samples. Legend has been fixed.

15. L360. ICI needs to be defined.

Ice Cloud Imager. The acronym has been spelled out now. Thank you for point that out.

16. Figure 9. The SD is very hard to see. color bar is missing.

I'm sorry but Fig. 9 are line plots so colorbars are not needed. And what does "SD" stand for?

RC3:

Unfortunately, I believe the writing quality for this article falls below the level neces- sary for publication. Grammatical errors are frequent (e.g. "collocation cases... are averaged separately considering they locate in different weather regimes at L228), articles/prepositions are often missing or used inappropriately, and word choices are often inappropriate, both in the sense of conventional English (e.g., "the biggest blob of power" at L304) and in terms of formal writing (e.g., "more and more" throughout). Occasionally, it was enough of an issue that I found it difficult to understand the author's message. Please make sure that this paper is proofread more carefully before the next submission, with specific attention to formal word choices.

We highly appreciate Reviewer #3's in-depth review and constructive suggestions. The English/grammar errors pointed by this and other reviewers have been corrected. The revised draft has been carefully proofread by the 4th author, who is a native speaker. The missing colorbars have now been added back. Below are point-by-point responses (questions in black, and responses in blue).

Specific comments are below:

L40: It would be nice to have more detail here. I have never heard of Cloud Radiative Effect before, and I would like to know which microphysical properties you are referring to.

Cloud radiative effect is defined as "the amount of **radiative** energy that would return to space if there were no **clouds**, minus the amount that actually escapes with **clouds** present." (https://svs.gsfc.nasa.gov/30603#:~:text=A%20simple%20way%20to%20describe,actually%20escapes%20with%20clouds%20present.) According to this definition, we can find that the details are dependent on how accurately we can understand and simulate the radiative effect from cloud microphysical and macrophysical properties.

There was a brief summary of some of the highly related papers discussing about cloud microphysical property impacts on cloud radiative effect in one of our previous papers [Gong et al., 2017 in the reference list, first paragraph in the introduction section]. Some of the sentences are quoted here: *"They are a major modulator of Earth's radiation and thus play an important role in weather and climate changes (e.g., Hartmann et al., 1984; Raymond & Zeng, 2000; Waliser et al., 2009). Studies have shown that the cloud radiative effect (CRE) of the clouds strongly depends on physical details, such as cloud top height (Kiehl et al., 1994), thickness (Hartmann & Berry, 2017; Hong et al., 2016), overlaying situation (Hartmann et al., 2001), and microphysical properties (Liou et al., 2002; Tang et al., 2017; Zeng et al., 2009a, 2009b). Fu and Liou (1993), for example, showed that the radiative heating rate for a layer of ice cloud with a fixed ice water path could differ by a factor of 10 when the mean effective radius of the ice particles varies by a factor of 5. Reducing the uncertainty of CRE requires a better understanding of ice cloud microphysical properties, which is essential not only to remote sensing of the bulk optical properties but also to the simulation of CRE (Tang et al., 2017)."*

L17/87: PD is defined in two different ways (Polarimetric radiance Difference, Polarimetric Difference). I would suggest unifying these.

Sorry for the confusion and thanks for point that out. We now clarified in the revised manuscript that "polarimetric difference" is an abbreviation of "polarimetric radiance difference".

L163: Has the collocated GMI, DPR, and CPR dataset been used by "many" other researchers for published articles? Yin et al. (2017) was the only one I was familiar with before reading this paper.

We also added another two references: Gong et al., 2017; Zeng et al., 2019. Also, "many" has been changed to "some". This is a really nice dataset that tremendously reduced the amount of work for us. The version I used did not include the ECMWF-AUX, but the latest version contained all collocated CloudSat and GPM Level 2 data products as well.

L193: ". Compared with "high-PD" scenario, the "low-PD" one apparently has more high clouds that are thinner than those from deep convective scenes as the reflectivity magnitudes are smaller". Unsure of what you mean by "more high clouds that are thinner". According to the CloudSat CFAD, the distribution of reflectivities for clouds above 13 km appear to extend larger at low PD than high PD, so I feel like that would make the high clouds at low-PD thicker?

Apologize for not being very clear here. What we meant was clouds at 15 km for comparison, which are mostly cirri. These cirri are thicker (higher reflectivities, likely from reminiscent from deep convective core or anvils) in the "deep convective" scenario than those in the "low-PD" scenario (reminiscent or in-situ formation). But now after rereading the sentences, we found such a different feature was not quite relevant to PD signals as neither GMI nor DPR could "see" cirrus cloud. So this sentence has been removed in the revised paper.

L203: I don't understand how a mean could be "PDF-weighted". Consider rewording?

Sorry we didn't realize that "PDF-weighted mean" was not a mathematically predefined term. Now it's changed to "weighted-mean".

L203-L218: I don't agree with the interpretations of reflectivity used throughout this paragraph, and I would prefer if the authors would be more objective. Reflectivity is used to describe the "thickness" of clouds and the "presence" of clouds in the same sentence, even though these two traits are not directly comparable. Later it is used to describe the number of precipitation sized particles (less/more), even though reflectivity can also be indicative of precipitation particle size. I would prefer if the figures are discussed more literally in this section, just explaining which PD scenarios have larger average reflectivity at different altitudes, and then save the interpretation of reflectivities for the following paragraph.

We agree with the reviewer that we put too many assertions in this paragraph which require many assumptions. In general, a single frequency radar reflectivity can only directly tell us about the mass at a given level, while larger mass could be associated with more particles, or larger particles (change of PSD), or denser particles (change of density). Here are the changes that have been made now in the revision:

Change "lesser present" to "thinner", and change "more and more thick" to "thicker". Here thinner and thicker correspond to the mass change according the weighted mean of reflectivity from different scenarios.

Change "suggesting" to "which might implicate", and change "indicates" to "might indicate", so the tone is weaken. This is our "hypothesis" that will be further elaborated using three frequency radar signals together, so we feel it necessary to raise up the idea step by step rather than just abruptly bringing it up after presenting all evidences.

Figure 3: I do not see any novel information provided by Ka band reflectivity in Figure 3b, so I think it could be removed.

We feel it's still worthwhile to keep Fig. 3b here, because our interpretation of GMI PD signals are mostly based on the triple-frequency radar reflectivity similarities and differences. The fact that Fig. 3b looks very similar to 3c but they both are quite different from Fig 3a indicate that 166 GHz PD signals are dominated by precipitation-sized particles, not cloud-sized particles.

On a similar note, Ka band reflectivity can be used to construct a Ku/Ka DWR. DWR can be used to provide information on ice particle size without being influenced by particle concentration, unlike the Z measurements used throughout the study. Comparison between DWR and Ku may also provide information on ice particle concentration. Considering the frequent discussion on ice particle aggregation, I think it could be valuable and relatively straightforward, to make figures similar to Figure 2 and 3 (and potentially Figure 8) with DPR-measured DWR. Keep in mind that DWR may also be influenced by liquid water attenuation, so a DWR profile should be interpreted with caution. This is more of a suggestion than a perceived requirement for publication, however.

As shown in Fig. 6 and Fig. 7c and 7d, DWR is not only a function of particle size, but also impacted a lot by density and multiple-scattering (also discussed in Battaglia et al., 2015), as well as liquid water attenuation as you mentioned here. Since Fig. 2 and 3 are in Section 3 that was intended for quantitative presentation of observations rather than discussions in Section 4, we think presenting Z instead of DWR will be more straightforward. Nevertheless, below is DWR_Ku/Ka for Fig. 3. One can see the red line is consistently smaller than the other three lines above the melting level.

[Figure]

Battaglia, A., S. Tanelli, K. Mroz, and F. Tridon (2015), Multiple scattering in observations of the GPM dual-frequency precipitation radar: Evidence and impact on retrievals, doi:10.1002/2014JD022866.

Figure 6: I am having difficulty understanding this figure well enough to determine whether I agree with your interpretation. The combination of filled and unfilled contours worked in Figure 2, but only because the zones of reflectivity were mostly separate. I think this figure would work better if both high PD and low PD were line contour plots of different colors, and regardless of whether that suggestion works, I would appreciate if this figure could be replotted to be more decipherable.

Now the colorbars have been added with rainbow color contours for "high-PD" and hue filled colors for "low-PD" scenarios. Hopefully now the contrasts between the two scenarios are clearer to see. Thanks for your suggestion.

RC4:

Unfortunately, the paper is lacking significantly in terms of grammar and phrasing. Some sentences are difficult to read and understand. Examples are found under grammar/technical comments below. Careful proofreading is required before publication.

We highly appreciate Reviewer #4's in-depth review and constructive suggestions, especially on providing so many detailed suggestions on English/grammar, which are really helpful. The English/grammar errors pointed by this and other reviewers have been corrected. The revised draft has been carefully proofread by the $4_{th}$ author, who is a native speaker. The missing colorbars have now been added back. Below are point-by-point responses (questions in black, and responses in blue).

Major comments:

1. While theoretical calculations are not the focus of the paper, only meant to augment the analysis, I think they could be described in more detail. For instance, what particle size distribution was assumed in the triple frequency calculations? More information can be given on the habits and the assumptions made for the riming model. This should not need more than two paragraphs I believe.

A new sub-section 2.4 is added to summarize the two RTMs and set-ups.

2. There is a lack of discussion regarding the sample size of the used data. Naturally, the amount of collocated GPM and CloudSat measurements of relevant cloud types are limited outside the polar regions. However, there is a lack discussion on how this could affect the analysis. There are only 62 high PD samples in total (table 1), meaning that features visible for the low PD data might not be captured for high PD data (in figure 6 for instance). How does this affect the credibility of the conclusions made in this study?

We admit that the original threshold for "high-PD" scene resulted an imbalanced sample pool size compared with "mid-PD" and "low-PD" scenarios, especially when collocation with CloudSat is further required. This caveat is now explicitly discussed in Section 2.3, which reads as "*Admittedly the sample size is strongly imbalanced between high-PD and low-PD scenes, as this is a trade-off between distinct disparities and statistical significance. Differences presented in Section 3 have passed the 95% statistical significance level unless otherwise noticed. But discussions in Section 4 are largely qualitative only*".

Specific comments:

1. L132: Could you please provide some motivation or background for the regime limits, especially the 150 K limit for deep convection.

This question was also raised by RC#2. Here is our response:

We agree with the reviewer that there is a counter-intuitive logic here in our definition. Usually people define "deep convection" based on the maximum radar reflectivity passing a certain threshold, which is more or less arbitrary as well, and the definition between CloudSat and DPR are not consistent because they work at different frequencies. Our definition is purely based on a 166 GHz TB threshold (TB < 150 K). This is of course arbitrary too.

If you revisit left panels in Fig.A2, you can see 150K corresponds to the very deepest depression of 166 GHz TB, which is the center of the deep convective line. Note that to make Fig. A2, we don't set up any

threshold but simply assign the coldest TB at each scan as the center of convective core. At 89 GHz this deep convective core ensemble is roughly < 225K. This value has been used previously in literatures to identify deep convections from TRMM TMI 85 GHz (e.g., Spencer et al., 1989; Nesbitt et al., 2000). So our 166 GHz threshold is consistent with 85 GHz threshold that studies used before. These two citations are now included to support our regime definition.

2. L198: "Because the KuPR reflectivity does not saturate with particle size as rapidly as Cloudsat, we can also infer large ice particles high in the atmosphere in the deep convective and low PD cases...". Do you mean that the limited sensitivity of KuPR to smaller ice hydrometeors indicates that high altitude KuPR reflectivities are due to large particles? Perhaps this could be phrased better.

Yes. This is what I mean. Now the sentence has been re-phrased as "Because the KuPR reflectivity is barely sensitive to cloud ice-sized particles, we can…".

3. L209: Could you perhaps use some other word than regime? Regime is already used (high-PD, deep convective, etc), so this caused some confusion for me. Perhaps "mode" is better?

Changed to "mode". Thank you for this suggestion.

4. L215: Please rephrase the sentence. Suggestion: "The latter scenario indicates the late stage of a convection life cycle, where the convective cell disappears and a stable stratiform layer forms to dominates the whole column."

Rephrased according to your suggestion. Thanks.

5. L262: What types of observations are referred to here? Reference?

Houze [2004] is a very nice review paper of MCSs. It cited a work by Bartels and Maddox [1991] where they collocated satellite IR observations with ground radiosonde observations to study the impact from low-level shear to the initiation and growth of MCSs. It also cited Kingsmill and Houze [1999] of TOGA CORE aircraft measurements of wind and MCSs.

6. L283: Should not the unit of the reflectivity ratio be unitless (or dBZ)? Confusing.

dBZ is decibel of relative to reflectivity, which is unitless. For example, see https://en.wikipedia.org/wiki/DBZ_(meteorology)

7. L294: I think it is prudent to also refer to Toyoshima (2015), for the DPR thresholds (Yin et al. (2017) added the CPR threshold).

Thank you. Citation added.

8. L294: Forgive me if I've missed this, but I can't find any description on the graupel spheroid in Leinonen and Szyrmer (2015). I suggest you provide at least a short description of both the habits and also on how the riming is modelled. I also find it contradictory to talk about a graupel spheroid under no-riming conditions, since graupel implies growth by riming.

Section 2.4 has been added to summarize the two RTM simulation set-ups, including PSD assumptions and particle model.

9. L296: Why is repositioning of the theoretical curves necessary? To augment the analysis? Please explain.

Because all theoretical curves always start from [0, 0] because naturally all DFR disappears if there's no water mass (equivalent LWP = 0). However, the origin from satellite triple-frequency diagram is centered at [5, 0], which is consistent with Yin et al. [2017]. Both of us believe this is likely due to the imperfect match between CloudSat and DPR. Therefore, theoretical curves are moved to [5, 0].

10. L339: How is density defined in this context?

Now section 2.4 has been added to summarize the two radiative transfer simulations. The second one used for generating isolines of Fig. 7 is a theoretical calculation of DFR as a function density, where snow, rain and mixed-phase particles were all assumed to be spherical, so density is indeed as simple as particle density.

11. L344: Sentence is very long and difficult to read.

Now this sentence has been broken apart into several short sentences. Hopefully now it's easier to read. Thanks for point that out. It's funny to re-read this sentence and realize how long it is.

12. L354: "evidences" -> "evidence". However, given the limitations (including sample size), I wonder if it would not be more prudent to use "indications" instead. I don't think this would take away from the novelty and importance of this work.

Thanks. Changed to "indications".

13. Section 6: I'm of the opinion that this section could better summarize the key conclusions and future work, in a more concise way. As it is now, it is rather a summary of what was presented and discussed in the paper. What are the take-home messages of this article?

This is not a short paper. Therefore in consideration of readers that do not have time to go through the paper, section 6 is a comprehensive summary of the key methods, outcomes and implications.

14. Figure 3: Could this figure include standard deviations (in dashed lines for instance)?

For simplicity and not overwhelm the lines with overlapping shades (as some lines are really close to each other), we intend to keep Fig. 3 as is. Thanks for your understanding.

15. Figure 5: I found it surprising that high PD signals are prevalent even in the presence of high wind speeds. Shouldn't high wind speeds promote random orientation? Just a comment.

Our interpretation is based on the conceptual picture of formation and boosting of MCS systems, where low-level wind shear was believed to really help the formation and fast development of MCS systems. Please see the last paragraph of Section 3.2 for our interpretations and references. Because ECMWF-AUX is interpolated from 0.5 X 0.5 degree, 3-hourly forecast data, there is no capability of resolving circulations within the cumulus convection, which is largely unresolved and parameterized. Therefore,

wind contrasts shown in Fig. 5 are more large-scale or meso-scale. For the high-level shear such like Fig. 5a, we really have no clear clue or interpretation at this moment.

Grammar/technical comments:

1. L14: "ambient environment" -> "the ambient environment"

Added. Thanks!

2. L15: "...impact up to the future climate projection and down to the details of the surface precipitation". Needs rephrasing. Suggestion: "...have on impact on climate projections as well as on the details of surface precipitation".

Suggestion adopted. Thanks.

3. L18: Remove "are".

Suggestion adopted. Thanks.
4. L43: "cloud" -> "clouds"

Suggestion adopted. Thanks.
5. L55: "tornado" -> "tornados"

Suggestion adopted. Thanks.

6. L72: Sentence needs rephrasing. Suggestion: "With swath widths typically over 1000 km and footprint sizes of 7-15 km, their combined usage can readily generate ice hydrometer production temporal and spatial scales that suits the needs of both weather and climate studies."

Suggestion adopted. Thanks.

7. L77: Problematic sentence. Suggestion: "While some of the recent products have advanced from using spherical ice models to more realistic habits, random orientation is still nearly always assumed, as it reduces computational complexity and decreases the degree of freedom for the otherwise severely under-constrained inversion problem."

This sentence has been rewritten according to RC#1's suggestion. It reads now as "While some of the recent products have advanced from using spherical ice models to more realistic habits [e.g., MODIS collection 6 assumed a bulk column-aggregate shape globally for its ice cloud properties retrieval, Platnick et al., 2017], random orientation is still nearly always assumed to avoid the complexity of deriving size and orientation simultaneously, as well as to avoid solving equations for 4 Stokes parameters simultaneously."

8. L81: "putting" -> "put"

Error corrected. Thanks.

9. L162: Sentence difficult to interpret. Should it be: "For example, Yin et al. [2017] used collocated CPR-DPR reflectivity profiles from this dataset to study the discrepancies found in triple frequency radar signatures and inferred different microphysical processes in convective and stratiform regimes."?

Suggestion adopted. Thanks.

10. L245: Sentence difficult to difficult. Please rephrase.

The situations are discussed under "in-cloud" and "ambient" conditions. Now it's rephrased as "Inside cloud, this feature …"

11. L253: It is difficult to understand what "which" refers to in the previous clause. Please rephrase for more clarity.

"Which" has been replaced with "Both of them".

12. L297: "property" → "properties"

Suggestion adopted. Thanks.

13. L303: Try use a more formal word than "blob". Perhaps "accumulation"?

Changed to "enhancement".

14. L356: "signal" -> "signals"

Suggestion adopted. Thanks.

15. L365: Problematic sentence. Suggestion: "It is well established that anvil clouds are likely associated with low-PD signals, while high-PD signals are instead linked to stratiform layers."

Suggestion adopted. Thanks.

16. L400: I think fig. 10 is introduced a bit suddenly here.

"The other way to evaluate the $PR_{sfc} - PD$ relationship is to composite the statistics." has been included to make the transition smoother.

17. L415: Sentence is difficult to understand. Please rephrase.

For squall line, "other precipitation flag" usually occur at the peripheries of the system. An example is given in the squall line cases below (from our ongoing work). Yellow is convective, light blue is stratiform, dark red is "other", and dark blue is no-precip. Although we think "other precipitation flag" should likely come from anvils, we need to make sure we are comparing trailing edge and leading edge to be consistent with the rest of this section. Therefore, an additional condition is added to exclude those "other" pixels that are at the trailing edge.

[Figure]

Among currently available spaceborne high-frequency microwave sensors, the GPM Microwave Imager (GPM-GMI) has a unique vertically-polarized (V-pol) and horizontally-polarized (H-pol) channel pair at 166 GHz. Gong and Wu [2017] found that the magnitude of 166 GHz polarization difference (PD), defined as the brightness temperature (TB) difference between V-pol and H-pol ($PD \equiv TB_V - TB_H$), is a good indicator of the presence of oriented ice particles. The largest PDs are found in moderately cold TB ($\sim$200K), corresponding to predominately horizontally oriented ice or snow particles inside medium thick ice cloud (e.g., anvils) or stratiform precipitation layer. This feature was also identified from 85 GHz TMI (Tropical Rainfall Measuring Mission's Microwave Imager) measurements [Prigent et al., 2005] and 157 GHz MADRAS (Megha-Tropiques's Microwave Analysis and Detection of Rain and Atmospheric Structures) measurements [Defer et al., 2014]. In particular, Olson et al. [2011] used TMI 89 GHz PD as one of the several parameters for stratiform/convective precipitation classification. PD approaches zero for clear-sky and deep convective cores. For the former, this is because 166 GHz is not sensitive to surface polarization when column water vapor exceeds abut 20mm [Zeng et al., 2019, Munchak et al., 2020]. As for the latter, Gong and Wu [2017] provided several possible explanations, including random orientation of ice particles induced by turbulent environment inside deep convective cores, large irregular-shaped graupel, or both V-pol and H-pol reach saturation at the same optical depth. Gong et al. [2017] further found that PD has a strong diurnal cycle over tropical land that is opposite to the diurnal cycle of cloud thickness and surface precipitation rate. The diurnal cycle of PD leads the latter two by $\sim$ 2 hours, indicating that ice microphysics change over the convection life cycle, which is important to the final precipitation

received at the ground. Nevertheless, all of the aforementioned papers studied passive sensor signals only. Scattering signals from passive sensors have very limited information on the vertical distribution of ice particles, and hence did not answer some

140 of the fundamental questions: **which altitude does PD information come from? What microphysical and environmental factors affect the observed PD variation over time and space? Can PD give more information in a broader context rather than just microphysics?** In this paper, these questions will be addressed by utilizing collocated GMI, DPR and CloudSat radar measurements as well as auxiliary environment variables.

This paper is organized as follows. Section 2 will introduce the dataset and methodology we use to make the composites of

145 climatology. We will present in Section 3 the differences of radar reflectivity, temperature and water vapor between high- and low-PD scenes. In Section 4, we will thoroughly discuss the underlying physical and microphysical mechanisms as well as consequences of such discrepancies. In Section 5, an ensemble of 46 squall line cases will be presented to showcase the potential use of high-frequency passive microwave PD observations to differentiate precipitation system life stage. Section 6 summarizes the whole work and points out several future study directions.

150 **2. Datasets and methodology**

**2.1 GPM core satellite and definition of PD regimes**

The Global Precipitation Measurement (GPM) mission core satellite, launched on February 27, 2014, carries the Dual-Frequency Precipitation Radar (DPR) and the GPM Microwave Imager (GMI). The GPM core satellite flies at an altitude of 407 km in a precessing orbit covering the Earth's $65°S$ to $65°N$. DPR is composed of a Ku-band radar (KuPR) and a Ka-band

155 radar (KaPR), making measurement at 13.6 GHz and 35.5 GHz, respectively. DPR scans cross-track with a footprint size of ~ $5 \times 5\ km^2$ at nadir and a swath width of 245 km for KuPR and 120 km for KaPR, respectively. Both KuPR and KaPR shoot 49 beams in each scan with a range resolution of 250 m (over-sampled to 125 m), but 25 KaPR beams are matched with KuPR footprints for the dual-frequency algorithm to work, and the remaining 24 beams are in interlaced mode with range resolution of 250 m. Therefore, there are total three modes of DPR scanning pattern: normal scan by KuPR (NS), matched scan by KaPR

160 (MS) and high-resolution interlaced scan by KaPR (HS)[1]. In this paper, we will mainly use KuPR measurements, and the central-25 MS measurement is used whenever "KaPR" is mentioned. The 2A.GPM.DPR Version 05A measured reflectivity without attenuation correction is used in this study. More mission details can be found at Skofronick-Jackson et al. [2018; 2019] and GPM website at https://gpm.nasa.gov/.

GMI is a 13-channel conical-scan microwave radiometer that sweeps the forward-looking cone at 48.5° (Earth incident angle

165 of 52.8°) from 10-89 GHz, and at 45° (Earth incident angle of 49.2°) from 166 to 183 GHz. Only the 166 GHz V-pol and H-pol measured brightness temperature (1B.GPM.GMI, Version 05A) will be considered in the current paper. The 166 GHz
* * *
[1] Prior to March 2018, the remaining 24 KaPR beams were interlaced at reduced vertical resolution but higher sensitivity to provide improved spatial sampling, but have since been matched to the outer swath KuPR to provide dual-frequency retrievals in the full DPR swath.

footprint size is $7.2 \times 4.2\ km^2$ (cross-track and along-track, respectively), and at this frequency the swath width is 885km on the Earth's surface in the cross-track direction with 221 pixels in each scan, the center part of which overlays with DPR scan during each GMI scan (https://pmm.nasa.gov/gpm/flight-project/gmi).

180 Gong and Wu [2017] constructed the two-dimensional probability density function (PDF) for PD-TBv relationship for different latitude ranges, one example is shown in Fig. 1 for the deep tropics ($5°S - Equator$). PD has a large spread when TB is in the middle of the observed range, implying different cloud and precipitation regimes are likely embedded in this moderately cold TB regime, which would be impossible to separate if TBv is the only metric to consider. For simplicity, we arbitrarily define four regimes in Fig. 1: Regime #1 ($TB < 150K, PD < 5\ K$) represents deep convective scenes (called "Deep

185 Convective Regime" hereafter); Regime #2, #3 and #4 share the same TB bounds ($150K < TB < 230K$), but different PD ranges, namely "low-PD" ($PD < 5\ K$), "medium-PD" ($5K < PD < 15\ K$) and "high-PD" ($PD > 15\ K$) regimes. Of course, these thresholds are arbitrary, but those thresholds based on TBs are consistent with previous literature. For example, $TB_{166GHz} < 150K$ is roughly equivalent to $TB_{85GHz} < 220K$ that was previously employed by Nesbitt et al. [2000]. This paper will focus on the differences between "low-PD" and "high-PD" regimes, as one can imagine that the situations falling

190 in "medium-PD" regime must be in transition status between the "low-PD" and "high-PD" scenarios. Since the general characteristics of the PD-TBv relationship is largely latitude-independent [Gong and Wu, 2014], this four-regime definition can be applied globally to all GMI measurements, except there are much fewer observations falling into the deep convective regime at high latitudes. Besides, shallow convection that is not as thick as deep convective cloud (e.g., congestus) may be wrongly classified into the "low-PD" regime. However, as the total congestus area is much smaller than anvil/stratiform

195 precipitation areas [e.g., Zeng et al., 2018], the immixture of shallow convection structures should have negligible impact on the general statistics.

**2.2 CloudSat radar and auxiliary datasets**

The CloudSat mission, launched on April 28, 2006 to a 705 km altitude Sun-synchronized orbit, carries the Cloud Profiling Radar (CPR). CPR is a nadir-looking W-band (94 GHz) radar with range resolution of 240 m and footprint size of

200 $1.4 \times 1.7 km^2$. The measured reflectivity vertical profiles from 2B-GEOPROF Version R05 product is used in this study.

As radar frequency increases from Ku-, Ka- to W-band, the radar sensitivity window also switches from precipitation to cloud. The CPR reflectivities are subject to strong attenuation from rain and multiple scattering from large precipitation particles. This becomes a serious issue in the range bins filled with heavy precipitation (i.e., from the melting layer to the ground). Due to complicated melting process within the melting layer, which often shows as a layer of enhancement of radar reflectivity (so-

205 called "bright band"), as well as the liquid attenuation issue, we will avoid discussing any reflectivity signals below 5 km (rough height of melting layer in the tropics) for all three radars throughout the paper. Water vapor throughout the profile can also attenuate the reflectivity signal by up to 5 dBZ for CPR [Marchand and Mace, 2018], but we still use measured reflectivity to avoid introducing additional assumptions that might complicate our analysis. The impact of water vapor attenuation at W-band will be touched upon later in the discussion.

**Commented [MSJ(2)]:** 67 pixels? Each GMI scan takes about 2 seconds and DPR takes less than a second, so I don't think this is what you meant…

ECMWF-AUX Version R05 dataset produced by the CloudSat team provides us auxiliary meteorological fields that are spatially and temporally interpolated to CloudSat range resolution volumes from ECMWF half-degree 3-hourly forecast data [Cronk and Partain, 2017]. Temperature, water vapor and horizontal wind profiles are compared for different PD scenarios.

**2.3 Collocation of radar and passive imager footprints – match and mismatch**

CloudSat-GPM Coincidence dataset Version 3B is a collection of collocated and coincident GMI, DPR and CPR measurements, which can be conveniently used for our current study. Details of collocation criteria and procedures can be found in Turk [2017]. This dataset has been used by some other researchers (e.g., Gong et al., 2017; Zeng et al., 2019). In particular, Yin et al. [2017] used collocated CPR-DPR reflectivity profiles from this dataset to study discrepancies found in triple frequency radar signatures and the inferred different microphysics processes in convective and stratiform regimes. In our study, we used more than three years of data (March 2014 – October 2017) to produce a total of 3040 coincident observations globally. This number of samples is based on GMI footprint; as DPR and CPR footprint sizes are smaller, we first averaged multiple DPR and CPR profiles to one collocated GMI footprint, and then group the averaged reflectivity, temperature, water vapor, zonal and meridional wind profiles into four regimes according to the PD-TBv values. Sample sizes separated in different categories can be found in Table 1. Admittedly, the sample size is strongly imbalanced between high-PD and low-PD scenes, as this is a trade-off between distinct disparities and statistical significance. Differences presented in Section 3 have passed the 95% statistical significance level unless otherwise noticed, but discussions in Section 4 are largely qualitative only.

Imperfect matching due to differences in footprint size, view-angle (i.e., atmospheric volume along line-of-sight is different even when the sight lines intersect), time or other factors can distort the compiled statistics. In our case, footprint size and line of sight mismatch are likely the largest sources of bias/uncertainty due to imperfect match. On one hand, the CPR footprint is much smaller than DPR's and GMI's footprints, and therefore, any cloud/precipitation inhomogeneity in the scale smaller than ~5 km can result in discrepancies that are hard to evaluate. On the other hand, since match-up is defined to happen whenever CPR beam intercepts with DPR beam at any altitude and at any DPR view-angle, the line-of-sight volume is quite different when DPR is at an off-nadir view-angle, and this problem is even more severe for GMI which always views at a slant angle. Even though a cosine function is multiplied to slightly mitigate this issue [Turk, 2017], 3D cloud inhomogeneity and beam-filling effects are again the culprit of uncertainty that is hard to justify. These two problems, however, are expected to be not too serious for our current study, because cloud inhomogeneity inside anvil and stratiform clouds is not as large as in deep convective scenes [Kirstetter et al., 2014]. Nevertheless, we know they will increase the uncertainty of our results, and temporal difference (allowable to be up to 15 minutes) has a similar impact. Only footprint mismatch might add an extra bias though, as will be discussed in Section 3.1.

As this coincident dataset does not contain collocated wind and bright band information from DPR, collocated indices are matched back to CloudSat ECMWF-AUX and 2A.GPM.DPR data files to extract the wind and bright band height/width information.
* * *
**2.4 Radiative transfer model simulations**

270 To help identify the microphysical property distinctions from the observed radar reflectivities and their differences, one radiative transfer model (RTM) and another theoretical calculation are employed. The first RTM and the simulation set-up are described in detail in Leinonen and Szyrmer [2015]. An aggregation model [Leinonen, 2013] was used to generate volumetric 3-D dry and heavily rimed dendrite aggregates because of their common occurrence in the atmosphere and aggregate efficiently. The particle size distribution (PSD) follows an inverse exponential distribution. The scattering computations for

275 equivalent spheroidal snowflakes were performed using T-matrix method (TM) based on Mishchenko and Travis [1998] and Self-Similar Rayleigh-Gan theory (SSRG) method based on Hogan and Westbrook [2014]. Using these two computational methods, the dynamic range of triple-frequency diagram, which will be in Section 4, can be largely covered based on many ground observations (e.g., Kneifel et al., 2015, 2016; Kulie et al., 2014; Neto et al., 2019).

The second radiative transfer theoretical calculation is employed to study the density impact on the radar signal difference,

280 which can help us diagnose which types of hydrometers likely dominate the signals under different scenarios. This model is described in detail in Liao and Meneghini [2011]. In particular, the snow follows the Gunn-Marshall size distribution [Gunn and Marshall, 1958], and rain follows the Marshall-Palmer size distribution [Marshall and Palmer, 1948]. Density and effective diameter follow a power-law form. All particles are assumed to be spheres. Simulations were verified before against airborne campaign data.

285 **3. Differences between high-PD and low-PD scenes**

**3.1 Radar reflectivity differences between high-PD and low-PD scenes**

Using 3.5 years of collocated radar reflectivity profiles, we can composite the two-dimensional probability density function (2D-PDF) respectively from CloudSat (color shaded) and KuPR (color contours) for the four regimes for the tropics, which is shown in Fig. 2. CloudSat's 2D-PDF separates "deep convective" scenario clearly from the rest three scenarios by having no

290 bright band kink at ~ 5 km, great amount of high clouds, and the center of highest occurrence of reflectivity located in the middle-upper troposphere (7-12 km) at around 15 dBZ. The PDF of "Low-PD" scenario is the closest to that of the "deep convective" scenario among the remaining three. As PD becomes larger, the bright band kink at ~ 5 km becomes more and more distinguished while the maximum occurrence of reflectivity also shifts down toward middle troposphere (5-8 km). This indicates the scene is more and more stratiform precipitation-like when PD magnitude increases. For KuPR's 2D-PDF, as Ku-

295 band is only sensitive to the precipitation-sized particles, we basically observe the same story as with CloudSat's 2D-PDF, except KuPR cannot see high altitude anvil clouds. Because the KuPR reflectivity is barely sensitive to cloud ice-sized particles, we can also infer large ice particles high in the atmosphere in the deep convective and low PD cases, while the strong increase in reflectivity towards the bright band in the high-PD case is indicative of aggregates.

The 1D plots of mean reflectivity profile from CloudSat, KaPR and KuPR ingeminate the preceding story in a more clear and

300 concise way, as shown in Fig. 3. Basically, Fig. 3 is the weighted mean along the x-axis of Fig. 2. Since the 2A.GPM.DPR

dataset also reports the altitude of the bright band (i.e., melting layer), Fig. 3b and 3c are plotted against altitude with respect to the melting level. As we stated in Section 2, we do not intend to discuss any signals below the melting layer since CloudSat reflectivity is likely strongly attenuated below the melting layer, and measured reflectivity is used for all three radars without any attenuation or multiple scattering correction. Above the melting layer, high-level cloud (> 9 km) is thinner while middle level cloud is thicker (5-8 km) when cloud regime switches from #1 "deep convective" (dark blue) to #4 "high-PD" (red). If we check the KaPR and KuPR profiles in Fig. 3b and 3c, however, we see roughly two distinct modes: one includes scenario#1 and #2 ("deep convective" and "low PD") that has more precipitation-sized particles throughout the upper-middle troposphere, which might imply that the convection and related cloud are still actively present within the column. On the contrary, the other mode, including scenario #3 and #4 ("medium" and "high" PDs), consist of fewer precipitation-sized particles aloft until close to the top of the melting layer due to increased temperature, where the sharp enhancement of reflectivity indicates fast and efficient growth from small ice particles to large fluffy snow aggregates. This is reasonable to happen microphysically because the sticking efficiency of two-ice-crystal collision increases rapidly near the melting layer. The latter mode might indicate the late stage of a convection life cycle, where the convective cell disappears and a stable stratiform layer forms to dominates the whole column.

Based on Fig. 2 and Fig. 3, we can summarize the discrepancies between high- and low-PD scenarios based on pure single-frequency radar observations: "low-PD" scenario has more high cloud and large ice particles high into the troposphere, implying active convective updrafts still in the development stage, while the "high-PD" scenario has much less high cloud but more middle-level cloud, with snow aggregation evident near the top of the melting layer. Therefore, "high-PD" scenario shows a distinct bright band, or melting layer signature, which is more "stratiform"-like and common in the decaying stage of convection. However, these composites are just snapshots, and without actually tracking the entire life cycle of convective system(s), these arguments need further validation. We will show some supportive evidence in Section 5 using an ensemble of squall lines. Furthermore, at this point, we cannot yet determine whether 
[revised manuscript text omitted]

Chase, R. J., J. A. Finlon, P. Borque, G. M. McFarquhar, S. W. Nesbitt, S. Tanelli, O. O. Sy, S. L. Durden, M. R. Poellot (2018), Evaluation of triple-frequency radar retrieval of snowfall properties using coincident airborne in situ observations during OLYMPEX, Geophys. Res. Letts., doi:10.1029/2018GL077997.

Chen, Q., J. Fan, S. Hagos, W. Gustafson Jr., L. K. Berg (2015), Roles of wind shear at different vertical levels: cloud system organization and properties, J. Geophys. Res. – Atm., doi:10.1002/2015JD023253.

Cronk, H. and P. Partain (2017), CloudSat ECMWF-AUX Auxillary data product process description and interface control document, http://www.cloudsat.cira.colostate.edu/sites/default/files/products/files/ECMWF-AUX_PDICD.P_R05.rev0_.pdf.

[revised manuscript text omitted]

835    Earth Sys. Sci. Data, doi:10.5281/zenodo.1341389.

Nesbitt, S. W., E. J. Zipser, D. J. Cecil (2000), A Census of Precipitation Features in the Tropics Using TRMM: Radar, Ice Scattering, and Lightning Observations, J. Clim., https://doi.org/10.1175/1520-0442(2000)013<4087:ACOPFI>2.0.CO;2

Olson, W. S., Y. Hong, C. D. Kummerow, and J. Turk (2001), A texture-polarization method for estimating convective-stratiform precipitation area coverage from passive microwave radiometer data, J. Appl. Meteor. and Clim., doi:10.1175/1520-

840    0450(2001)040<1577:ATPMFE>2.0.CO;2

Platnick, S. et al. (2017), The MODIS Cloud Optical and Microphysical Products: Collection 6 Updates and Examples From Terra and Aqua, in IEEE Transactions on Geoscience and Remote Sensing, vol. 55, no. 1, pp. 502-525, doi: 10.1109/TGRS.2016.2610522.

[revised manuscript text omitted]

---

## Author Response (AR2)

Response to editor and reviewers' suggestions:

We are asked to enhance the figure resolution. We didn't realize that "save as" button provided by office word had two options for generating PDF documents, one of which is to print as online version that contains lower resolution figures.

Now the updated PDF contains higher resolution figures better suited for print-out. In addition, we've uploaded all original figures in a zip file at "supplement" section. Every figure has 300 dpi and at least 8 cm for the shortest length, strictly following the ACP publication requirements. The only exception is Fig. 1, as it is adapted and modified from one of our previous papers, so the figure size is kept as the original one. The subplots in Fig. 4 and 5 are rearranged from vertical to horizontal layouts to ease the final typesetting. Blue and red lines in Fig. 9c have been switched in color to be consistent with the color definition in Fig. 9a and 9b to avoid confusions.

In addition, the "acknowledgement" section has been modified to acknowledge the three anonymous reviewers and Dr. Toshi Masui's insightful comments, which make this presentation better and clearer to read. We also include funding acknowledgement for two co-authors in this section. "Author contribution" section is also modified slightly to reflect Dr. Munchak's effort on polishing the final English writings.